# Vibrator and PI4KIIIα govern neuroblast polarity by anchoring non-muscle myosin II

Chwee Tat Koe[1], Ye Sing Tan[1], Max Lönnfors[2], Seong Kwon Hur[2], Christine Siok Lan Low[3], Yingjie Zhang[1,4], Pakorn Kanchanawong[3,5], Vytas A Bankaitis[2], Hongyan Wang[1,4,6]*

[1]Neuroscience and Behavioral Disorders Programme, Duke-NUS Medical School, Singapore, Singapore; [2]Department of Molecular and Cellular Medicine, Texas A&M University Health Science Center, College Station, United States; [3]Mechanobiology Institute, National University of Singapore, Singapore, Singapore; [4]NUS Graduate School for Integrative Sciences and Engineering, National University of Singapore, Singapore, Singapore; [5]Department of Biomedical Engineering, National University of Singapore, Singapore, Singapore; [6]Department of Physiology, Yong Loo Lin School of Medicine, National University of Singapore, Singapore, Singapore

**Abstract** A central feature of most stem cells is the ability to self-renew and undergo differentiation via asymmetric division. However, during asymmetric division the role of phosphatidylinositol (PI) lipids and their regulators is not well established. Here, we show that the sole type I PI transfer protein, Vibrator, controls asymmetric division of *Drosophila* neural stem cells (NSCs) by physically anchoring myosin II regulatory light chain, Sqh, to the NSC cortex. Depletion of *vib* or disruption of its lipid binding and transfer activities disrupts NSC polarity. We propose that Vib stimulates PI4KIIIα to promote synthesis of a plasma membrane pool of phosphatidylinositol 4-phosphate [PI(4)P] that, in turn, binds and anchors myosin to the NSC cortex. Remarkably, Sqh also binds to PI(4)P in vitro and both Vib and Sqh mediate plasma membrane localization of PI(4)P in NSCs. Thus, reciprocal regulation between Myosin and PI(4)P likely governs asymmetric division of NSCs.

DOI: https://doi.org/10.7554/eLife.33555.001

*For correspondence:
hongyan.wang@duke-nus.edu.sg

**Competing interests:** The authors declare that no competing interests exist.

## Introduction

Understanding how neural stem cells divide asymmetrically is central for stem cell and cancer biology. *Drosophila* neural stem cells (NSC), or neuroblasts, are an excellent model for understanding stem cell asymmetry and homeostasis. Each neuroblast divides asymmetrically to generate a self-renewing neuroblast and a differentiating daughter cell. It is the latter that ultimately produces neurons/glia (*Doe, 2008*). Disruption of asymmetric division in neuroblasts may result in the formation of ectopic neuroblasts, leading to brain tumor development (*Caussinus and Gonzalez, 2005*; *Lee et al., 2006*; *Wang et al., 2006*; *Wang et al., 2007*; *Chabu and Doe, 2009*; *Wang et al., 2009*; *Chang et al., 2010*; *Wang et al., 2011*). Asymmetric division of neuroblasts depends on the polarized distribution of proteins and their asymmetric segregation into different daughter cells. Apical proteins such as atypical protein kinase C (aPKC), Par-6 (Partitioning defective 6), Par-3/Bazooka (Baz), Inscuteable (Insc) and Partner of Inscuteable (Pins) control the localization of basal proteins, as well as the orientation of the mitotic spindle (*Knoblich, 2010*; *Chang et al., 2012*). Basal cell fate determinants Numb, Prospero (Pros), Brain tumor (Brat), and their adaptor proteins Partner of Numb (Pon) and Miranda (Mira) are critical for neuronal differentiation upon asymmetric segregation

**eLife digest** Stem cells are cells that can both make copies of themselves and make new cells of various types. They can either divide symmetrically to produce two identical new cells, or they can divide asymmetrically to produce two different cells. Asymmetric division happens because the two new cells contain different molecules. Stem cells drive asymmetric division by moving key molecules to one end of the cell before they divide.

Asymmetric division is key to how neural stem cells produce new brain cells. Many studies have used the developing brain of the fruit fly *Drosophila melanogaster* to understand this process. Errors in asymmetric division can lead to too many stem cells or not enough brain cells. This can contribute to brain tumors and other neurological disorders. Fat molecules called phosphatidylinositol lipids are some of chemicals that cause asymmetry in neural stem cells. Yet, it is not clear how these lipid molecules affect cell behavior to turn stem cells into brain cells.

The production of phosphatidylinositol lipids involves proteins called Vibrator and PI4KIIIα. Koe et al. examined the role of these two proteins in asymmetric cell division of neural stem cells in fruit flies. The results show that Vibrator activates PI4KIIIα, which leads to high levels of a phosphatidylinositol lipid called PI(4)P within the cell. These lipids act as an anchor for a group of proteins called myosin, part of the machinery that physically divides the cell. Hence, myosin and phosphatidylinositol lipids together control asymmetric division of neural stem cells.

Further experiments used mouse proteins to compensate for defects in the equivalent fly proteins. The results suggest that the same mechanisms are likely to hold true in mammalian brains, although this still needs to be proven. Nevertheless, given that human equivalents of Vibrator and PI4KIIIα are associated with neurodegenerative disorders, schizophrenia or cancers, these new findings are likely to help scientists better to understand several human diseases.

DOI: https://doi.org/10.7554/eLife.33555.002

into the differentiating daughter cell (*Gonzalez, 2007*; *Doe, 2008*; *Knoblich, 2010*). Asymmetric localization of Mira was thought to be achieved via a series of linear inhibitory regulations from aPKC to Mira via Lethal giant larvae (Lgl) and nonmuscle Myosin II (*Kalmes et al., 1996*; *Barros et al., 2003*; *Betschinger et al., 2003*). aPKC was shown later to directly phosphorylate both Numb and Mira to polarize them in neuroblasts (*Smith et al., 2007*; *Atwood and Prehoda, 2009*), while Lgl directly inhibits aPKC in neuroblasts, rather than displacing Mira on the cortex (*Atwood and Prehoda, 2009*). However, how myosin regulates asymmetric division of neuroblasts remains not well understood.

Despite having identified phosphatidylinositol (PI) lipids as critical components of cellular membranes important for cell polarity of various cell types (*Krahn and Wodarz, 2012*), the role of PI lipids and their regulators are not well established in *Drosophila* neuroblasts. Phosphoinositides are phosphorylated derivatives of PI, and their synthesis is catalyzed by lipid kinases; specially, phosphoinositide 4-phosphate [PI(4)P] production is catalyzed by PI 4-OH kinases (PI4K). Interestingly, these lipid kinases are inherently inefficient enzymes and production of biologically sufficient amounts of PI(4)P for efficient signaling requires the stimulation of these enzymes by phosphatidylinositol transfer proteins (PITPs). PITPs leverage their lipid exchange activities to present PI as a superior substrate to PI4Ks (*Bankaitis et al., 2010*; *Grabon et al., 2015*). Classical PITPs bind to PI and phosphatidylcholine (PC) in a mutually exclusive manner, with PI being the preferred binding substrate (*Wirtz, 1991*). Classical PITPs are also divided into two different subtypes, type I and type II, both of which contain PITP domains with type II PITPs containing additional N-termini of large membrane-associated modules (*Nile et al., 2010*). Loss of PITP function is associated with neurological disorders, including cerebellar ataxia (*Hsuan and Cockcroft, 2001*; *Cockcroft and Garner, 2011*). These deficits are consistent with the fact that PITPs are particularly highly expressed in the brain and cerebellum (*Utsunomiya et al., 1997*). In mammals, type I PITPs consist of PITPα and PITPβ. Mouse hypomorphic mutants for PITPα/Vibrator exhibit a progressive, and ultimately fatal, whole-body tremor that reflects neurodegenerative disease, whereas mice with the PITPα structural gene deleted survive to term but die shortly after birth (*Hamilton et al., 1997*; *Alb et al., 2003*). Attempted rescue of PITPα-null mice with a PITPα mutant specifically defective in PI-binding exhibit phenotypes

indistinguishable from those of the null mutant (*Alb et al., 2007*). In *Drosophila*, a single type I PITP has been identified, named Vibrator/Giotto, which plays a prominent role in cytokinesis of spermatocytes and neuroblasts (*Gatt and Glover, 2006*; *Giansanti et al., 2006*). However, the role of PITPs in neuroblast polarity is unknown.

Herein, we show that the type I PITP Vibrator regulates asymmetric division of neuroblasts through anchoring nonmuscle myosin II regulatory light chain (RLC), Sqh, to the cell cortex. We further demonstrate that depletion of PI4KIIIα(a PI4 kinase that is essential for PI(4)P synthesis) also leads to Sqh delocalization and asymmetric division defects. Thus, Vibrator and PI4KIIIα likely promote synthesis of a plasma membrane PI(4)P pool that, in turn, binds and anchors myosin to the cell cortex. Importantly, we show that Sqh is able to bind to phospholipid PI(4)P and facilitates its membrane localization in neuroblasts. Taken together, the data identify a reciprocal dependence between Myosin and PI(4)P localization in neuroblasts.

## Results

### Vib controls homeostasis of larval central brain neuroblasts

To identify novel regulators of neuroblast homeostasis, we performed a genetic screen on a collection of chromosome 3R mutants induced by ethyl methane sulfonate (EMS) mutagenesis (CT Koe, F Yu and H Wang, unpublished). We identified two independent alleles of *vibrator*, which were designated as *vibrator133* (*vib133*) and *vibrator1105* (*vib1105*). While the mutation(s) in *vib1105* is/are yet to be identified, *vib133* contains a mutation (AG->TG) at the splice acceptor site of its last intron (*Figure 1A*), presumably causing a premature stop codon. Hemizygous *vib133* or *vib1105* over deficiency *Df(3R)BSC850* survived to 2nd instar larval stage, similar to hemizygous *vibj7A3*, a reported protein null allele, over a deficiency *Df(3R)Dl-BX12* (*Gatt and Glover, 2006*). Both hemizygotes, *vib133*/*Df(3R)BSC850* and *vib1105*/*Df(3R)BSC850*, die earlier than hemizygous *vibj5A6*, a P-element insertion allele, over *Df(3R)BSC850*, which survived to 3rd instar larval stage. Similar to the Vib protein levels in *vibj7A3*/*Df(3R)BSC850* (12%), Vib protein levels with the predicted size 32 KDa in the larval brains of hemizygous *vib133*/*Df(3R)BSC850* and *vib1105*/*Df(3R)BSC850* were both significantly reduced to 9% compared with control (*Figure 1—figure supplement 1A*, n = 2). The detection of trace amounts of Vib in *vibj7A3*/*Df(3R)BSC850* was likely due to the higher sensitivity of the new anti-Vib antibody we generated compared with previously reported antibody. Truncated Vib protein with predicted size of 26 KD was not detected in *vib133*/*vibj5A6* protein extracts (*Figure 1—figure supplement 1A*), suggesting that the truncated form might be unstable. Taken together, these results suggest that *vib133* and *vib1105* are most likely strong hypomorphic alleles.

To analyze neuroblast homeostasis in these *vib* alleles, we generated MARCM clones of both type I and type II neuroblasts (*Lee et al., 2000*). Type I wild-type control clones always contain one neuroblast that expresses pan-neural genes Deadpan (Dpn) and Asense (Ase) (*Figure 1B*, n = 38), while type II clones possesses one neuroblast that expresses Dpn but not Ase (*Figure 1C*, n = 34). By contrast, we observed that 56% (n = 41) of type I *vib133* clones and 61% (n = 41) of type II *vib133* clones showed ectopic neuroblasts (*Figure 1B,C*). In addition, 28.7% (n = 115) *vib133* clones lost neuroblasts (*Figure 4—figure supplement 2A*). Similarly, 28.3% (n = 99) *vib1105* clones were devoid of neuroblasts, with ectopic neuroblasts observed in 20% (n = 35; data not shown) of type I *vib1105* clones and 50% (n = 36) of type II *vib1105* clones (*Figure 1B,C*).

Loss of neuroblasts upon Vib depletion is unlikely due to cell death, because active Caspase-3 is absent in both wild-type control and *vib133* neuroblasts (*Figure 1—figure supplement 2A*). Rather, it is likely partially due to premature differentiation of neuroblasts as none (n = 15) of type I neuroblast control clones expressed nuclear Pros, a differentiation factor, while 66.7% (n = 12) of type I *vib133* neuroblast clones exhibited nuclear Pros (*Figure 1—figure supplement 2B*). Furthermore, in *vib133/j5A6* and *vib1105/j5A6* mutants, 50% (n = 158) and 56.8% (n = 243) of interphase neuroblasts exhibited nuclear Pros expression, respectively (*Figure 1—figure supplement 2C*; control, no nuclear Pros, n = 121). Given that the nuclear Pros observed in *vib-* neuroblasts was very weak, we cannot exclude the possibility that other yet-to-be-identified factors are responsible for the premature differentiation of *vib-* neuroblasts. The average size of *vib133* neuroblasts (7.76 μm, n = 27) is smaller than neuroblasts from control MARCM clones (10.79 μm, n = 37) (*Figure 1—figure supplement 2D*). We measured the ratio of nucleolar to nuclear size in neuroblasts, which is an indicator of

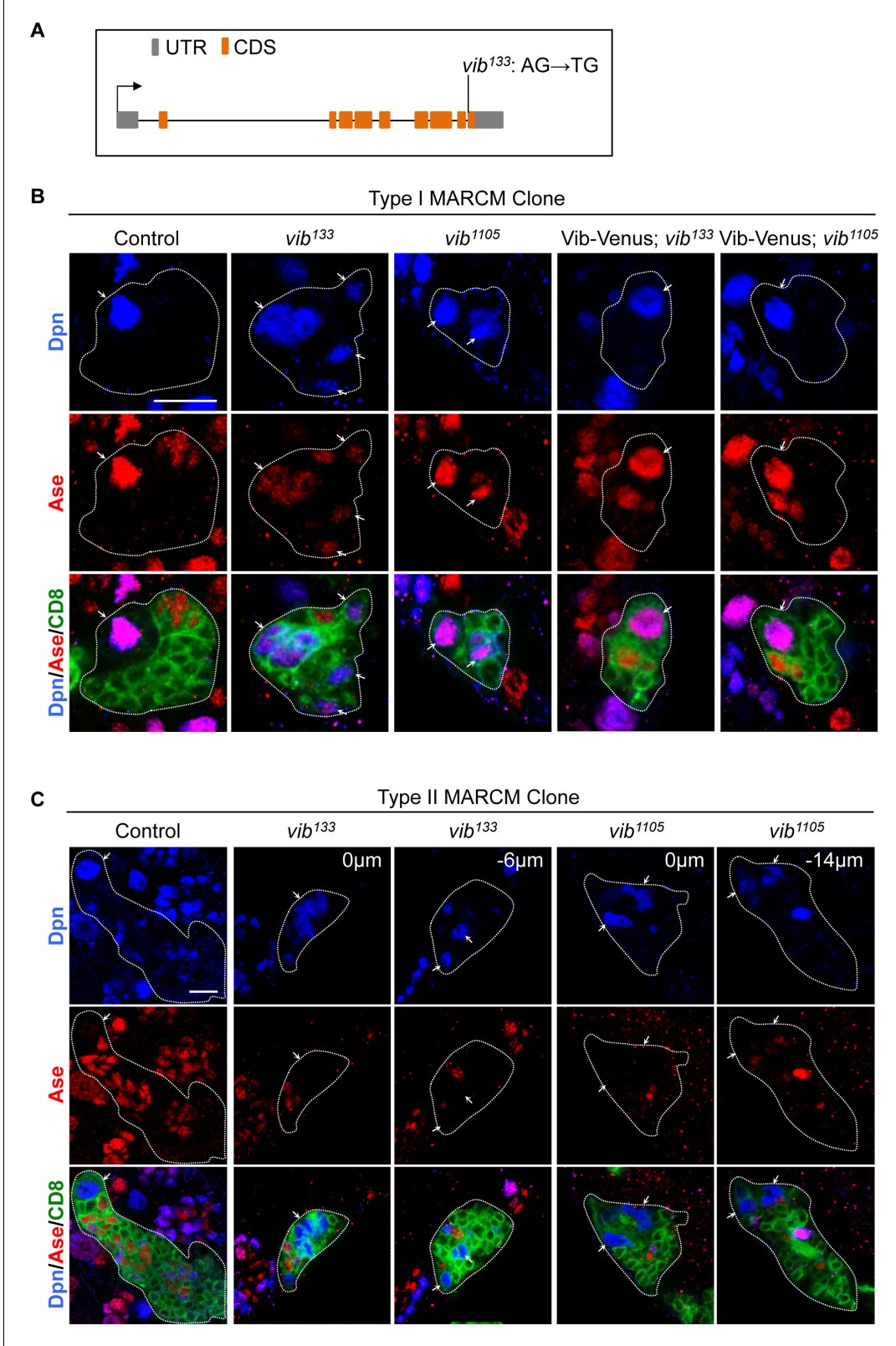

**Figure 1.** Vibrator regulates homeostasis of larval brain neuroblasts. (**A**) A schematic showing the mutation in *vib^133* allele. (**B**) Type I MARCM clones of control (FRT82B), *vib^133*, *vib^1105*, UAS-Vib-Venus; *vib^133* and UAS-Vib-Venus; *vib^1105* were labeled with Dpn (blue), Ase (red) and CD8::GFP (green). Note that the signal of Vib-Venus in MARCM clones was masked by intense CD8::GFP signal. (**C**) Type II MARCM clones of control (FRT82B), *vib^133* and

*Figure 1 continued on next page*

*Figure 1 continued*

$vib^{1105}$ were labeled with Dpn (blue), Ase (red) and CD8::GFP (green). Arrows indicate neuroblasts. Clones are marked by CD8::GFP and indicated by white dotted lines. Scale bars, 10 μm.

DOI: https://doi.org/10.7554/eLife.33555.003

The following figure supplements are available for figure 1:

**Figure supplement 1.** Mammalian PITPs rescue *vib* defects in neuroblast homeostasis.

DOI: https://doi.org/10.7554/eLife.33555.004

**Figure supplement 2.** Loss-of-neuroblast in *vib* mutants is due to premature differentiation, but not apoptosis.

DOI: https://doi.org/10.7554/eLife.33555.005

cellular growth. With nucleolus: nucleus ratio in wild-type control neuroblasts normalized as 1 (n = 29), this ration in $vib^{133}$ neuroblasts was reduced slightly to 0.81 (n = 28) (*Figure 1—figure supplement 2E*). This result suggest that cell growth in $vib^-$ mutant neuroblasts is mildly reduced, likely due to the nuclear localization of Pros resulting in earlier termination of proliferation as well as premature differentiation.

We also observed cytokinesis defects as previously reported in *vib* mutants (*Gatt and Glover, 2006*; *Giansanti et al., 2006*) (*Figure 4—figure supplement 1A* and data not shown). Our evidence suggest that ectopic neuroblasts are unlikely caused by cytokinesis defects (refer to page 13–14). Quantification of neuroblast growth was performed on those clones without obvious cytokinesis block. We fused wild-type Vib with Venus at the C-terminus and generated transgenic flies expressing Vib-Venus. Ectopic neuroblasts in either mutant *vib* alleles were well rescued by expressing Vib-Venus. Ectopic neuroblasts were rescued in 91.3% of type I $vib^{133}$ (n = 23), 100% of type II $vib^{133}$ (n = 14), 100% of type I $vib^{1105}$ (n = 46) and 96% of type II $vib^{1105}$ (n = 25) neuroblast lineages (*Figure 1B* and *Figure 1—figure supplement 1B*). Together, these results suggest that Vib regulates neuroblast homeostasis in both type I and type II neuroblast lineages.

By performing pairwise sequence alignment (EMBOSS Needle) between Vib with rat PITPs, we observed 59.6% and 61.8% identity between Vib with rat PITPα and PITPβ, respectively. Given this sequence homology, we next explored if mammalian PITP homologs could replace Vib function in *Drosophila* neuroblasts. To this end, we generated transgenic flies expressing either rat PITPα or PITPβ, which were each fused with Venus at the C-terminus. Expression of PITPα completely rescued ectopic neuroblasts in both type I (n = 43) and type II (n = 26) MARCM clones of $vib^{133}$ (*Figure 1—figure supplement 1C*). Likewise, expression of PITPβ in $vib^{133}$ fully rescued ectopic neuroblasts in both type I (n = 100) and type II (n = 39) MARCM clones (*Figure 1—figure supplement 1C*). These results suggest that mammalian PITPs may play a conserved role in neuroblast homeostasis.

Vib was previously reported to localize to both the cleavage furrow and spindle envelope in spermatocytes and only to the spindle envelope in neuroblasts (*Gatt and Glover, 2006*; *Giansanti et al., 2006*). Since the anti-Vib antibody did not work consistently in neuroblasts, we analyzed Vib-Venus expression under the control of *insc*-Gal4. Vib-Venus was strongly localized to the cell cortex and weakly to the spindle envelope throughout the cell division cycle (*Figure 1—figure supplement 1D*). In telophase neuroblasts, Vib-Venus was cortically localized and slightly enriched at the cleavage furrow (*Figure 1—figure supplements 1D*, 20%, n = 10). To determine if Vib-Venus is enriched at the basal side, we measured the pixel intensity of GFP at both the apical and basal cortex of neuroblasts. At metaphase and telophase, neuroblasts showed an average Venus$_{Basal}$: Venus$_{Apical}$ ratio of 1.26 (n = 32) and 1.39 (n = 33), respectively. In addition, the average ratio of Vib-Venus intensity at cleavage furrow to that at the apical cortex (termed Venus$_{CF}$: Venus$_{Apical}$) of telophase neuroblasts was 1.33 (n = 33) (*Figure 1—figure supplement 1F*). The slight enrichment of Venus-Vib at the basal side was likely due to the attachment of multiple GMCs surrounding the neuroblasts, which is commonly seen for membrane proteins.

## Vib is required for asymmetric division of neuroblasts

Given that neuroblast homeostasis is disrupted in *vib* mutants, we assessed if Vib is required for asymmetric division of neuroblasts. In wild-type control, metaphase neuroblasts localize Par complex, Insc and the Pins-Gαi complex to the apical cortex, while displacing Mira-Brat-Pros and Numb-Pon complexes to the basal cortex (*Figure 2A–D*, *Figure 2—figure supplement 1A–C*, and data

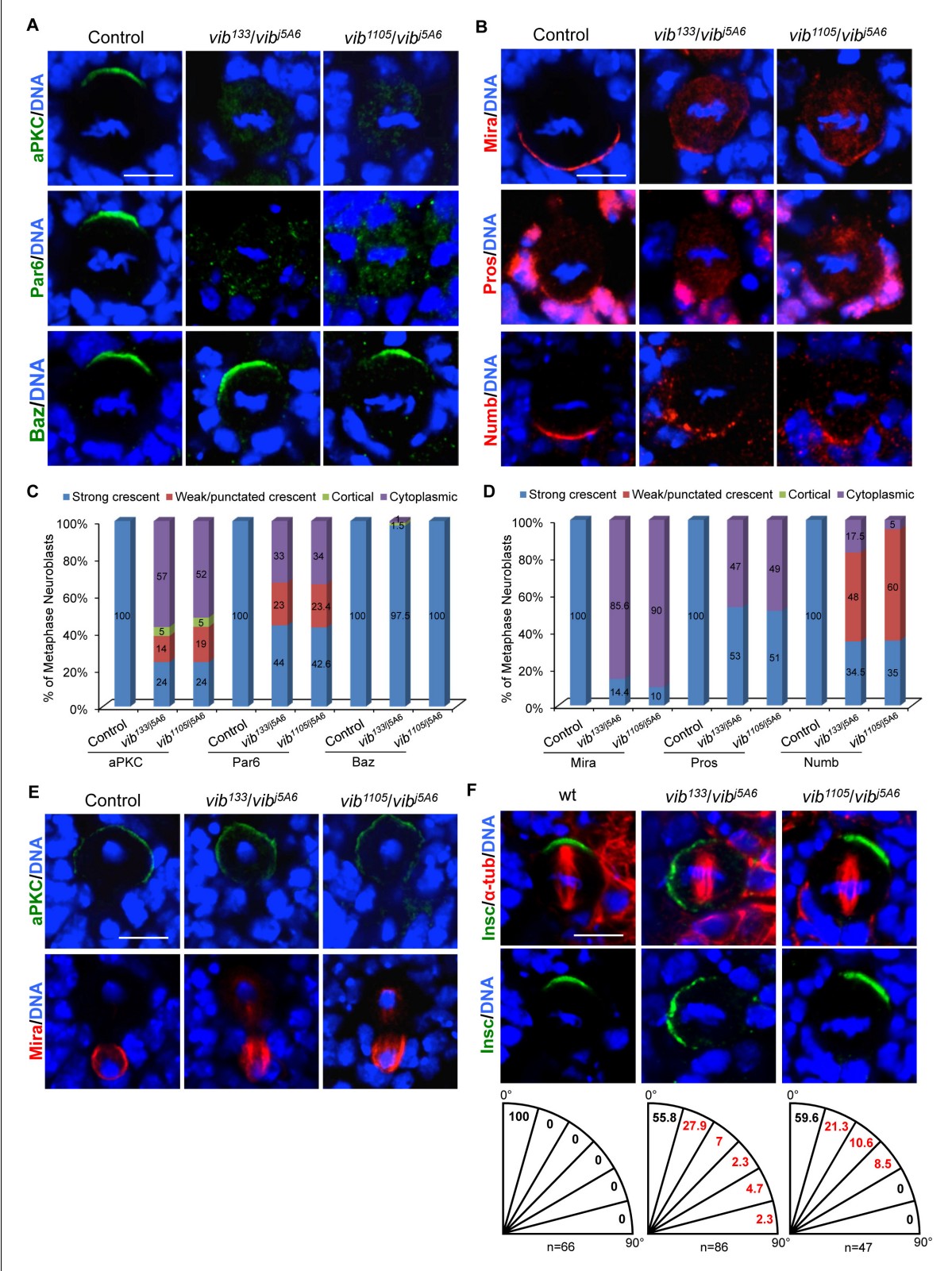

**Figure 2.** Vib regulates asymmetric cell division of neuroblasts. (A) Metaphase neuroblasts of wild-type control, *vib^133^/vib^j5A6^* and *vib^1105^/vib^j5A6^* were labeled with aPKC, Par-6 or Baz (green) and DNA (blue). (B) Metaphase neuroblasts of control, *vib^133^/vib^j5A6^* and *vib^1105^/vib^j5A6^* were labeled with Mira, Pros or Numb (red) and DNA (blue). (C) Quantification of aPKC, Par-6 and Baz localization in metaphase neuroblasts for (A). (D) Quantification of Mira, Pros and Numb localization in metaphase neuroblasts for (B). (E) Telophase neuroblasts of control, *vib^133^/vib^j5A6^* and *vib^1105^/vib^j5A6^* were labeled with

*Figure 2 continued on next page*

*Figure 2 continued*

aPKC (green), Mira (red) and DNA (blue). (F) Metaphase neuroblasts of wild-type control, *vib*[133]/*vib*[j5A6] and *vib*[1105]/*vib*[j5A6] were labeled with Insc (green), α-tubulin (red) and DNA (blue). Spindle orientation was determined by the angle between apical-basal axis inferred by the Insc crescent and the mitotic spindle axis indicated by α-tubulin. Quantification of spindle orientation in control, *vib*[133]/*vib*[j5A6] and *vib*[1105]/*vib*[j5A6] are shown below the images. Scale bars, 5 μm.

DOI: https://doi.org/10.7554/eLife.33555.006

The following figure supplement is available for figure 2:

**Figure supplement 1.** Vib is required for asymmetric protein localization in neuroblasts.

DOI: https://doi.org/10.7554/eLife.33555.007

not shown; 100%, n = 42). By contrast, in 76% of both *vib*[133/j5A6] and *vib*[1105/j5A6] metaphase neuroblasts, aPKC was delocalized from the apical cortex to the cytoplasm, and exhibited weak or punctate crescent profiles, or was uniformly distributed on the cell cortex (*Figure 2A,C*; n = 42 for both trans-heterozygotes). Consistent with the delocalization of aPKC, we also observed autophosphorylated aPKC (p-aPKC) dispersed from the apical cortex into the cytoplasm in *vib*[133/j5A6] (57%, n = 21) and *vib*[1105/j5A6] (53%, n = 32) metaphase neuroblasts (*Figure 2—figure supplement 1A,C*). Since aPKC is important for neuroblast self-renewal (*Rolls et al., 2003*), delocalization of aPKC into the cytoplasm is likely responsible for the loss-of-neuroblast phenotype observed in *vib* mutants. Similarly, 56% (n = 39) and 57.4% (n = 47) of metaphase neuroblasts with *vib*[133/j5A6] and *vib*[1105/j5A6], respectively, showed delocalization of Par-6 into the cytoplasm, or exhibited as weak/punctate crescent (*Figure 2A,C*).

Membrane localization of apical protein Baz is mediated by direct interaction of its C-terminal region with two phosphatidylinositol phosphates (PIPs), namely PI(3,4,5)P3 and PI(4,5)P2 (*Heo et al., 2006*; *Krahn et al., 2010*). Surprisingly, Baz localization remained unaffected in *vib*[133/j5A6] (*Figure 2A,C*, 97.5%, n = 68) and *vib*[1105/j5A6] (*Figure 2A,C*, 100%, n = 65) metaphase neuroblasts. This observation suggests that Vib controls localization of aPKC and Par-6 via a mechanism largely independent of Baz. An alternative possibility is that sufficient PI(4,5)P2 and PI(3,4,5)P3 remained in those *vib* mutant neuroblasts to support apical localization of Baz. Furthermore, Gαi was partially delocalized to the cytoplasm (*Figure 2—figure supplements 1A,C*, 18.6%, n = 59 for *vib*[133/j5A6] and 21.3%, n = 80 for *vib*[1105/j5A6]), while localization of Insc (n = 38 for *vib*[133/j5A6], n = 78 for *vib*[1105/j5A6]), Pins (n = 34 for *vib*[133/j5A6], n = 50 for *vib*[1105/j5A6]) and Cdc42 (n = 24, in *vib*[133/j5A6]) was largely unaffected (data not shown). Taken together, the data demonstrate that Vib is particularly important for the localization of aPKC and Par-6 in neuroblasts.

Next, we examined the localization of basal proteins in *vib* mutant neuroblasts. While Mira was asymmetrically localized in 100% of wild-type neuroblasts during metaphase (*Figure 2B,D*, n = 60), its localization was severely disrupted in metaphase neuroblasts of *vib*[133/j5A6] and *vib*[1105/j5A6]. It was either delocalized to the cytoplasm or mis-localized to the mitotic spindle (*Figure 2B,D*, *vib*[133/j5A6], 85.6%, n = 118; *vib*[1105/j5A6], 90.4%, n = 42). In wild-type neuroblasts, Pros and Brat, two cargo proteins of Mira, were basally localized (*Chang et al., 2012*) (*Figure 2B,D*, *Figure 2—figure supplements 1B,C*, 100%, n = 50). However, Pros (*Figure 2B,D*, *vib*[133/j5A6], 47%, n = 36; *vib*[1105/j5A6], 49%, n = 47) and Brat (*Figure 2—figure supplement 1B,C*, *vib*[133/j5A6], 42%, n = 102; and *vib*[1105/j5A6], 33.3%; n = 33) were delocalized to the cytoplasm or mitotic spindle in *vib* metaphase neuroblasts. In wild-type controls, cell fate determinant Numb is basally localized in metaphase neuroblasts (*Figure 2B,D*, n = 53). However, 65.5% (n = 63) of *vib*[133/j5A6] and 65% (n = 98) of *vib*[1105/j5A6] metaphase neuroblasts showed punctate crescent or cytoplasmic localization of Numb (*Figure 2B,D*). Furthermore, 30% (n = 10) and 22.5% (n = 80) of metaphase neuroblasts showed cytoplasmic localization of PON in *vib*[133/j5A6] and *vib*[133/j5A6], respectively (*Figure 2—figure supplement 1B,C*).

At telophase, mis-localized apical or basal proteins can be corrected and localize asymmetrically in a phenomenon named 'telophase rescue' (*Peng et al., 2000*). While apical localization of aPKC was largely restored at telophase in *vib* neuroblasts (*vib*[133/j5A6], 97.7%, n = 43; and *vib*[1105/j5A6], 100%, n = 23), Mira (94%; n = 101 for *vib*[133/j5A6], 95.6%; n = 23 for *vib*[1105/j5A6]), Pros (*Figure 4—figure supplements 2B,C*, 54.8%; n = 31 for *vib*[133/j5A6], 71.4%; n = 21 for *vib*[1105/j5A6]) and Brat (42%; n = 19 for *vib*[133/j5A6]) remained on the mitotic spindle, in addition to their basal localization (*Figure 2E* and data not shown). Consistently, 44% of metaphase neuroblasts of *vib*[j5A6]/*Df(3R) BSC850* hemizygotes showed cytoplasmic aPKC localization (*Figure 2—figure supplement 1D*,

n = 50) and 94% showed Mira delocalization (*Figure 2—figure supplement 1D*, n = 50). Mira remained mis-localized to the mitotic spindle in 85% of *vib$^{j5A6}$/Df(3R)BSC850* telophase neuroblasts (*Figure 2—figure supplement 1D*, n = 27). The ratio of segregated apical/basal proteins determines whether cells adopt neuroblast or GMC fate following the neuroblast division (*Cabernard and Doe, 2009*). Given that Mira/Pros/Brat was still observed at the basal side of the cortex, it is unclear whether the formation of ectopic neuroblasts is partially contributed by the reduced amount of basal determinants eventually segregated into basal daughter cell due to their mislocalization on the spindle. Taken together, we conclude that Vib is required for localization of apical and basal protein and their faithful segregation in dividing neuroblasts.

We ascertained whether spindle orientation was affected in mitotic neuroblasts of *vib* mutants. To this end, we measured the angle between apicobasal axis inferred by the Insc cortical crescent and mitotic spindle axis in metaphase neuroblasts. In wild-type control metaphase neuroblasts, the mitotic spindle is always parallel to the apicobasal polarity (*Figure 2F*, n = 66). By contrast, we observed that 44.2% of metaphase neuroblasts in *vib$^{133/j5A6}$* showed mis-orientation of the mitotic spindle axis with respect to neuroblast polarity (*Figure 2F*, n = 86). Likewise, 40.4% of metaphase neuroblasts in *vib$^{1105/j5A6}$* showed mitotic spindle mis-orientation (*Figure 2F*, n = 47). We observed orthogonal division in *vib$^{133/j5A6}$* metaphase neuroblasts (*Figure 2F*; 2.3%, n = 86), suggesting that spindle orientation defects likely contributed, at least partially, to the altered sibling cell fate observed in *vib* mutants. With the known function of Gαi in mitotic spindle orientation (*Schaefer et al., 2001*; *Yu et al., 2003*), partial delocalization of Gαi (*Figure 2—figure supplement 1A,C*) likely contributed to the spindle orientation phenotype in *vib* mutants. Centrosomal Mud localization was largely unaffected in *vib$^{133/j5A6}$* metaphase neuroblasts (*Figure 2—figure supplement 1E*; 96.3%, n = 27; Control, n = 23). Centrosomes in *vib$^{133/j5A6}$* neuroblasts were unaffected, as Cnn localization is normal in both control (n = 24) and *vib$^{133/j5A6}$* (n = 47) neuroblasts (*Figure 2—figure supplement 1F*). Likewise, the spindle architecture is unaffected in *vib$^{133/j5A6}$* (*Figure 2—figure supplement 1G*; n = 43; Control, n = 25). They were able to assemble normal-looking mitotic spindle and astral microtubules that were labeled by α-tubulin. Taken together, Gαi delocalization, but not spindle architecture or centrosomal defects, most likely caused the spindle orientation defects observed in *vib* mutants.

## Vib anchors non-muscle myosin II regulatory light chain Spaghetti-squash (Sqh) to the cell cortex in neuroblasts

The non-muscle myosin II regulatory light chain protein Spaghetti-squash (Sqh) regulates both asymmetric localization of basal cell fate determinants in neuroblasts and cytokinesis of dividing cells (*Karess et al., 1991*; *Barros et al., 2003*). In particular, Mira was delocalized to the mitotic spindle in *sqh$^-$* embryonic neuroblasts (*Barros et al., 2003*). In the null allele, *sqh$^{Ax3}$*, neuroblasts failed to divide in MARCM clones due to severe cell division defects, precluding us from analyzing their asymmetric division. To circumvent this problem, we knocked down *sqh* by RNAi under *insc*-Gal4. Upon *sqh* RNAi knockdown Mira was delocalized to the cytoplasm in larval neuroblasts (*Figure 3—figure supplements 1A*, 14.7%, n = 34). We also observed aPKC delocalization (*Figure 3—figure supplements 1A*, 16.2%, n = 37) in *sqh* RNAi neuroblasts. Our data confirmed the role of Sqh in regulating asymmetric division of larval brain neuroblasts.

We next investigated whether the localization of Sqh is dependent on Vib function. To this end, we examined localization of GFP-tagged Sqh under its endogenous promotor (Sqh::GFP) in *vib$^{133/j5A6}$* and *vib$^{1105/j5A6}$*. In wild-type larval brains, Sqh::GFP was largely cytoplasmic at interphase with 38% cortical localization (*Figure 3—figure supplements 1B*, 38%, n = 95), while localizing uniformly to the cortex of neuroblasts at metaphase (*Figure 3A,B*, 100%, n = 53). As mitosis progressed, its distribution was restricted to the cleavage furrow at telophase (*Cabernard et al., 2010*) (*Figure 3A, B*). However, in *vib$^{133/j5A6}$*, Sqh::GFP was observed in the cytoplasm in 92% of interphase neuroblasts (*Figure 3—figure supplement 1B*, n = 123) and 42.6% of metaphase neuroblasts (*Figure 3A, B*, n = 61). In 29.4% of *vib$^{133/j5A6}$* telophase neuroblasts, Sqh::GFP failed to be accumulated at the cleavage furrow and became uniformly cortical and cytoplasm (*Figure 3A,B*, n = 34). Likewise, in *vib$^{1105/j5A6}$*, cortical localization of Sqh::GFP was diminished in 46.8% of metaphase neuroblasts (*Figure 3A,B*, n = 79), while 24.2% of telophase neuroblasts had increased cytoplasmic localization of Sqh::GFP (*Figure 3A,B*, n = 33). Consistently, Sqh localization examined by Sqh antibody revealed

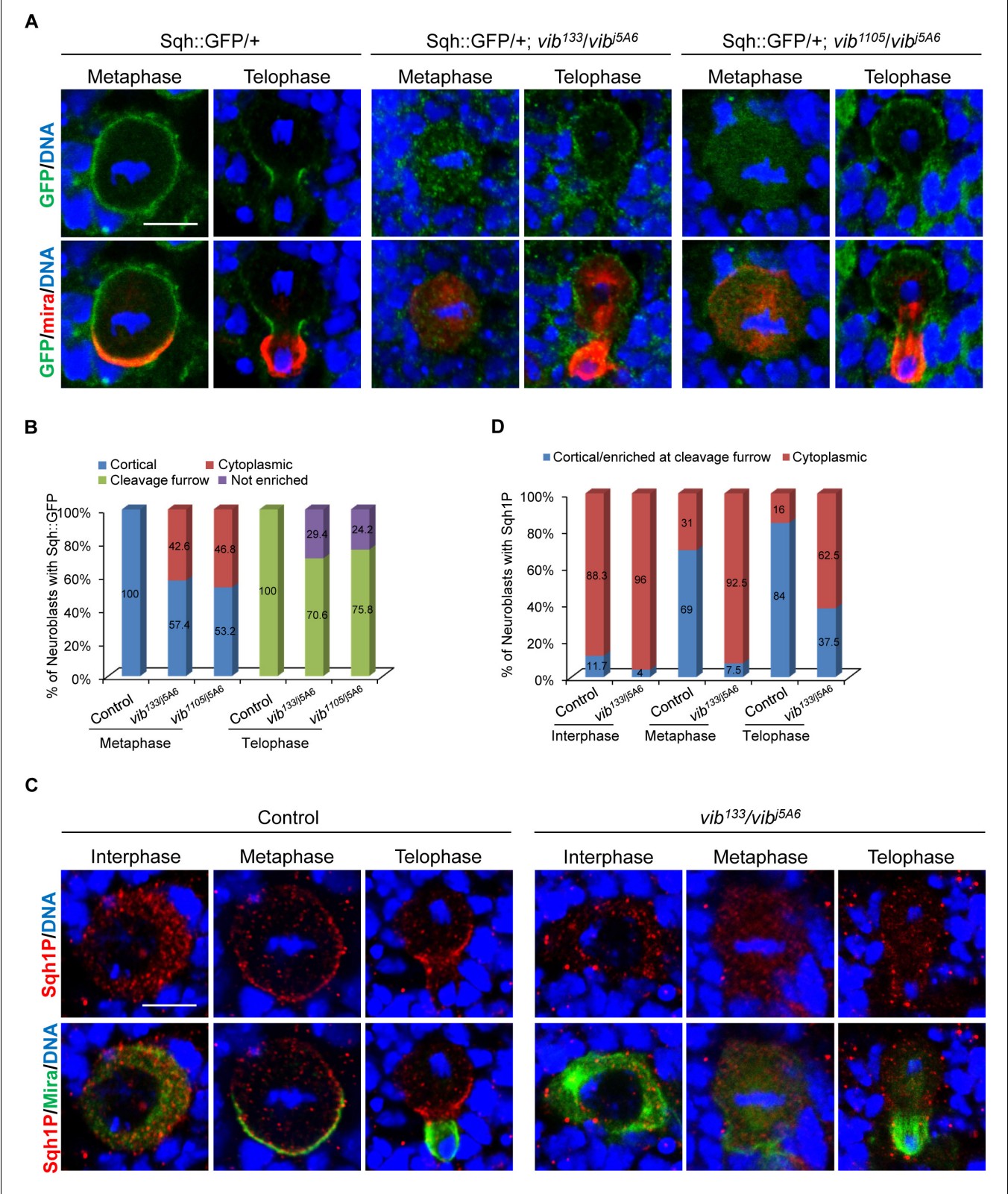

**Figure 3.** Vib anchors non-muscle myosin II regulatory light chain Sqh to neuroblast cortex. (**A**) Neuroblasts of control, $vib^{133}$/$vib^{j5A6}$ and $vib^{1105}$/$vib^{j5A6}$ each expressing one copy of Sqh::GFP were labeled with GFP (green), Mira (red) and DNA (blue). (**B**) Quantification of Sqh::GFP localization in neuroblasts for (**A**). 'Not enriched' in quantification legend refers to no enriched localization at cleavage furrow. (**C**) Neuroblasts of control and $vib^{133}$/$vib^{j5A6}$ were labeled with Mira (green), Sqh1P (red) and DNA (blue). (**D**) Quantification for Sqh1P localization in neuroblasts for (**C**). Scale bars, 5 μm.

*Figure 3 continued on next page*

*Figure 3 continued*

DOI: https://doi.org/10.7554/eLife.33555.008

The following figure supplement is available for figure 3:

**Figure supplement 1.** Sqh localization is disrupted in *vib* mutant neuroblasts.

DOI: https://doi.org/10.7554/eLife.33555.009

that its cortical localization at metaphase (*Figure 3—figure supplements 1C*, 75%, n = 24) was disrupted in $vib^{133/j5A6}$ neuroblasts.

Myosin activity is regulated by phosphorylation of myosin regulatory light chain (MRLC) (*Ikebe and Hartshorne, 1985*; *Ikebe et al., 1986*; *Ikebe et al., 1988*). *Drosophila* MRLC Sqh has two phosphorylated forms, monophosphorylated at Ser21 (Sqh1P) or diphosphorylated at both Thr20 and Ser21 (Sqh2P) (*Zhang and Ward, 2011*). In imaginal discs, Sqh1P largely localizes to the adherens junction, while Sqh2P localizes to the apical domain (*Zhang and Ward, 2011*). To determine if the localization of active form of Sqh is affected in *vib* mutant, we examined Sqh1P in $vib^{133/j5A6}$, as the Sqh2P antibody is no longer available. During interphase, similar to Sqh::GFP, Sqh1P was largely cytoplasmic in both control and $vib^{133/j5A6}$ neuroblasts (*Figure 3C,D*). In wild-type mitotic neuroblasts, Sqh1P was seen at both the cell cortex (69%, n = 52) and as diffuse cytoplasmic staining at metaphase (31%, n = 52), and enriched at the cleavage furrow at telophase (*Figure 3C,D*, 84%, n = 25). By contrast, Sqh1P was cytoplasmic in the vast majority of metaphase neuroblasts (92.5%, n = 40) and it failed to enrich at the cleavage furrow in telophase neuroblasts (62.5%, n = 24) of $vib^{133/j5A6}$ (*Figure 3C,D*). Taken together, these observations indicate that Vib is required for the localization of Sqh to the cell cortex in mitotic neuroblasts.

In neuroblasts apical Rho kinase (Rok) phosphorylates and activates Sqh, ensuring cleavage furrow positioning (*Tsankova et al., 2017*). To test whether Vib is necessary for the correct localization and activation of Rok, we first determined localization of Rok-GFP in *vib* mutant neuroblasts. In controls (Ubi-Rok-GFP heterozygous), neuroblasts expressed cortical Rok-GFP at metaphase (93.5%, n = 46) and showed enriched Rok-GFP at the cleavage furrow in telophase (91.7%, n = 24). Similarly, in $vib^{133/j5A6}$ we observed cortical Rok-GFP localization in 91.6% of metaphase neuroblasts (n = 24) and 85.7% (n = 14) of telophase neuroblasts showed enriched Rok-GFP at the cleavage furrow (*Figure 3—figure supplement 1D*). This result suggests that Vib is not essential for Rok localization in neuroblasts. Next, we explored whether increased Rok activity could rescue the asymmetric cell division defects observed in *vib* mutants. We overexpressed the catalytic domain of Rok (Rok-CAT), a previously characterized transgene (*Winter et al., 2001*) that is known to increase phosphorylation of Sqh, and found that it was unable to recuse Mira delocalization in $vib^{133/j5A6}$ (*Figure 3—figure supplement 1E*; metaphase, 85.7%, n = 42; telophase, 83.3%, n = 12), rather showing similar Mira delocalization to $vib^{133/j5A6}$ trans-heterozygous mutant (metaphase, 90.2%, n = 34; telophase, 84.6%, n = 26). Expression of RokCAT in wild-type neuroblasts had no influence on Mira localization in neuroblasts (*Figure 3—figure supplement 1E*; n = 27 for metaphase; n = 13 for telophase). These observations suggest that Vib regulates Sqh localization unlikely via localizing/activating Rok.

## The lipid binding and transfer activity of Vib is critical for asymmetric division and homeostasis of neuroblasts

Given that the signature biochemical activity of Vib is the heterotypic exchange of PI and PC, we investigated whether this lipid exchange activity is important for its role in regulating asymmetric division and homeostasis of neuroblasts. The lipid headgroup binding cavity of PITPs comprised of several conserved residues that are important for interaction with PI and PC (*Cockcroft and Garner, 2011*). Among these residues, mammalian Threonine 59, the Thr63 equivalent in *Drosophila* Vib, has been extensively studied (*Cockcroft and Garner, 2011*). The Thr59 to Ala substitution of mammalian PITPα reduces PI binding specifically, whereas the Glu substitution specifically ablates the PI binding/transfer activity without affecting PC binding/transfer activities (*Alb et al., 1995*; *Morgan et al., 2004*; *Cockcroft and Garner, 2011*). We purified VibT63A (Thr63 to Ala) and VibT63E (Thr63 to Glu) and determined their PI and PC binding and transfer capacities in vitro. In vitro PI and PC binding and transfer activities were measured by monitoring the PyrPtdCho or [3 hr]-PtdCho and PyrPtdIns or [3 hr]-PtdIns transport from donor to acceptor liposomes as previously

described (*Somerharju et al., 1987*; *Bankaitis et al., 1990*; *Schaaf et al., 2008*). Similar to rat PITPα and PITPβ, wild-type Vib protein bound both PC and PI (*Figure 4A–B*). In addition, Vib transferred both PI and PC robustly, albeit with slightly lower capacity than mouse PITPs and the major yeast PI/PC-transfer protein Sec14 (*Figure 4C,D*). By contrast, mutation of Thr63 to Ala abolished both PI and PC binding and transfer capacity (*Figure 4A–D*). While PI binding and transfer were abolished in VibT63E, the mutant protein retained reduced PC binding and transfer capability (*Figure 4A–D*). Similar to the mutant mammalian PITPα (*Tilley et al., 2004*; *Shadan et al., 2008*), WF to AA mutation at position 202 and 203 of Vib, which are situated in the membrane binding region of Vib abolished both PI and PC binding as well as their transfer (*Figure 4A–D*). Together, these data demonstrate that, similar to mammalian PITPα, *Drosophila* Vib processes PI and PC binding and transfer activity, with VibT63E including additional defects in PC binding and transfer.

We next determined the ability of VibT63A and VibT63E to rescue asymmetric division defects observed in *vib*[133/j5A6]. We generated transgenic flies expressing Venus tagged VibT63A (VibT63A::Venus) and Venus tagged VibT63E (VibT63E::Venus) in larval brains (*Figure 4—figure supplement 1A*). Expression of VibT63A::Venus alone in larval brain had no influence on Mira and aPKC localization in mitotic neuroblasts (*Figure 4E,F*, n = 44 and data not shown). Mira was still delocalized in 83.7% (*Figure 4E,F*, n = 43) of metaphase neuroblasts and 86.2% of telophase neuroblasts (*Figure 4E,F*, n = 29) with VibT63A::Venus expression in *vib*[133/j5A6] transheterozygotes, indistinguishable from *vib*[133/j5A6] control neuroblasts (*Figure 4E,F*, 88%, n = 69 for metaphase; 85.7%, n = 42 for telophase). Furthermore, expression of VibT63A::Venus also failed to rescue ectopic neuroblasts in type I (47.6%, n = 21) and type II MARCM clones (57.7%, n = 26) of *vib*[133] (*Figure 4—figure supplement 1B,C*). Similarly, expression of VibT63E::Venus in larval brains of *vib*[133/j5A6] still resulted in strong delocalization of Mira (*Figure 4E,F*, 91%, n = 45 for metaphase; 86.3%, n = 38 for telophase) and aPKC (*Figure 4—figure supplements 1D*, 51%, n = 90 for metaphase; telophase rescue, n = 47 for telophase). Furthermore, in *vib*[133/j5A6] transheterozygotes expressing VibT63E, 60.6% (n = 33) of metaphase neuroblasts showed cytoplasmic Pros localization, similar to the delocalization of Pros in *vib*[133/j5A6] metaphase neuroblasts (*Figure 4—figure supplement 2B,C*; 53.3%, n = 45). At telophase, 52.6% of *vib*[133/j5A6] neuroblasts expressing VibT63E showed Pros delocalization at the mitotic spindle (*Figure 4—figure supplement 2B,C*, n = 21), undistinguishable from *vib*[133/j5A6] telophase neuroblasts (*Figure 4—figure supplement 2B,C*; 54.8%, n = 31).

We next tested whether lipid binding/transfer activity of Vib is critical for Sqh localization in neuroblasts. In *vib*[133/j5A6] neuroblasts expressing VibT63A::Venus, 88.1% (n = 59) of metaphase neuroblasts and 71.4% (n = 21) of telophase neuroblasts showed delocalization of Sqh. Similarly, in *vib*[133/j5A6] neuroblasts expressing VibT63E::Venus, 88.4% of metaphase neuroblasts (n = 43) displayed cytoplasmic Sqh localization and 53.3% of telophase neuroblasts (n = 15) failed to enrich Sqh at the cleavage furrow (*Figure 4—figure supplement 3A,B*). These defects were similar to those observed in *vib*[133/j5A6] (*Figure 4—figure supplement 3A,B*). Likewise, Sqh1P delocalization observed in *vib*[133/j5A6] neuroblasts failed to be rescued by overexpressing either VibT63A or VibT63E (*Figure 4—figure supplement 3C,D*). At metaphase, 80% (n = 65) of *vib*[133/j5A6] neuroblasts expressing VibT63A::Venus were cytoplasmic and only 45% (n = 20) of telophase neuroblasts were localized at cleavage furrow. Likewise, 85.7% (n = 70) of *vib*[133/j5A6] neuroblasts expressing VibT63E::Venus displayed cytoplasmic Sqh1P at metaphase and only 38.5% (n = 26) of neuroblasts had Sqh1P localization at the cleavage furrow at telophase (*Figure 4—figure supplement 3C,D*). Taken together, these observations indicate that lipid binding and transfer activity of Vib is critical for its role in regulating asymmetric division of neuroblasts.

Vib has a previously known role in cytokinesis of spermatocytes and neuroblasts (*Gatt and Glover, 2006*; *Giansanti et al., 2006*). In MARCM clones, 78.6% of *vib*[133] neuroblasts displayed a cytokinesis defect (*Figure 4G*, *Figure 4—figure supplement 1B*, n = 70). Surprisingly, expression of VibT63E::Venus rescued the cytokinesis defect in 98.7% of *vib*[133] clones (n = 75), while excess neuroblasts or depletion of neuroblasts persisted (*Figure 4G*, *Figure 4—figure supplement 1B,C*, and *Figure 4—figure supplement 2A*). 24% of *vib*[133] MARCM clones expressing VibT63E were depleted of neuroblasts (*Figure 4—figure supplement 2A*, n = 99). Similar to *vib*[133/j5A6] neuroblasts (*Figure 4—figure supplements 2B,C*, 41.6%, n = 327), Pros was ectopically localized to the nucleus in interphase *vib*[133/j5A6] neuroblasts expressing VibT63E (*Figure 4—figure supplements 2B,C*, 41%, n = 315), suggesting that nuclear Pros might be partly contributed to premature differentiation and subsequent loss of neuroblasts in these mutant brains. By contrast, expression of VibT63A::Venus, in

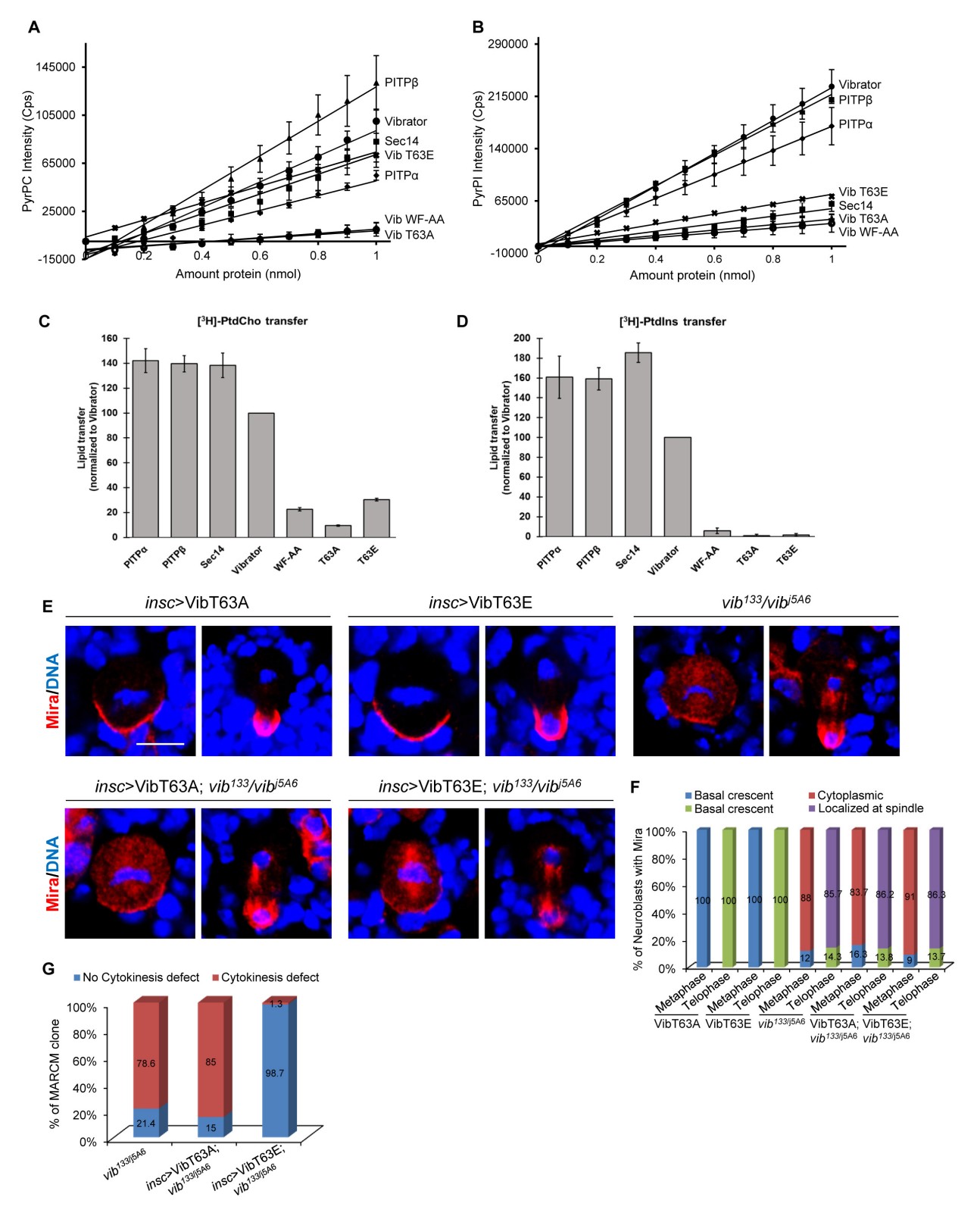

**Figure 4.** Lipid transfer activity of Vib is important for asymmetric cell division. (**A**) Fluorescence intensity measured at 378 nm of PyrPtdCho as a function of protein amount. An increase in intensity represents binding events when the fluorophore is removed from its quenched environment. Lines represent linear equation trend lines were shown. For the error bars, n = 2. (**B**) Fluorescence intensity measured at 378 nm of PyrPtdIns as a function of protein amount. An increase in intensity represents binding events when the fluorophore is removed from its quenched environment. Lines represent

*Figure 4 continued on next page*

*Figure 4 continued*

linear equation trend lines were shown. For the error bars, n = 2. (**C**) Endpoint PtdCho transfer assays. Indicated recombinant proteins (10 μg/assay) were assayed for [3 hr]-PtdCho transfer. Transfer efficiencies of Vibrator were set at 100% (n = 3), and transfer of all other proteins are normalized relative to Vibrator activity values obtained from the same experiments. (**D**) Endpoint PtdIns transfer assays. Indicated recombinant proteins (10 μg/assay) were assayed for [3 hr]-PtdIns. Transfer efficiencies of Vibrator were set at 100% (n = 3), and transfer of all other proteins are normalized relative to Vibrator activity values obtained from the same experiments. (**E**) Neuroblasts of *insc* >VibT63A, *insc* >VibT63E, *vib*$^{133}$/*vib*$^{j5A6}$, *insc* >VibT63A in *vib*$^{133}$/*vib*$^{j5A6}$ and *insc* >VibT63E in *vib*$^{133}$/*vib*$^{j5A6}$ were labeled with Mira (red) and DNA (blue). (**F**) Quantification for Mira localization in neuroblasts for (**E**). 'Localized at Spindle' refers to the delocalization of Pros at the mitotic spindle in addition to basal crescent in telophase neuroblasts. (**G**) Quantification showing the percentage of MARCM clones of *vib*$^{133/j5A6}$, *insc* >VibT63A; *vib*$^{133/j5A6}$ and *insc* >VibT63E; *vib*$^{133/j5A6}$ that exhibited cytokinesis defects. Scale bars, 5 μm.

DOI: https://doi.org/10.7554/eLife.33555.010

The following figure supplements are available for figure 4:

**Figure supplement 1.** Lipid transfer activity of Vib is required for neuroblast homeostasis.
DOI: https://doi.org/10.7554/eLife.33555.011
**Figure supplement 2.** Lipid binding and transfer activity of Vib is required for the localization of Pros in neuroblasts.
DOI: https://doi.org/10.7554/eLife.33555.012
**Figure supplement 3.** Lipid binding and transfer activity of Vib is required for the localization of Sqh in neuroblasts.
DOI: https://doi.org/10.7554/eLife.33555.013

which both PI and PC binding and transfer capacity were abolished, did not rescue the cytokinesis defects observed in *vib*$^{133}$ neuroblasts (85%, n = 27; data not shown). Since VibT63E retains partial PC binding and transfer activity (*Figure 4A–D*), overexpression of VibT63E shown in *Figure 4—figure supplement 1A* might restore sufficient PC binding and transfer activity to rescue cytokinesis defects in *vib* mutants. Thus, despite the pleotropic phenotypes observed in *vib* neuroblasts, asymmetric division defects are unlikely a consequence of cytokinesis failure as these two phenotypes could be uncoupled in the above experiments.

## Sqh binds to phosphoinositide PI(4)P and localizes it to cell cortex in neuroblasts

Given that Sqh is targeted to the neuroblast cortex by Vib, a lipid binding and transfer protein, we examined whether Sqh was able to bind to phospholipids. To perform lipid-binding assays, Myc-tagged Sqh was expressed in S2 cells (*Figure 5A*) and protein extracts were incubated with lipid strips. While the control did not bind to phosphoinositides, Myc-Sqh was able to bind to PI(4)P, PI(4,5)P$_2$, PI(3, 4, 5) P$_3$ and phosphatidylethanolamine (*Figure 5A*). Therefore, we provide the initial evidence that Sqh binds to PIs in vitro.

PITPs stimulate PI4K activity for PI(4)P production (*Cockcroft and Carvou, 2007*) (*Figure 5—figure supplement 1A*). In wild-type larval brain neuroblasts, PI(4)P was barely detected using an anti-PI(4)P antibody in wild-type larval brains (*Forrest et al., 2013*), presumably due to its low abundance. In addition, PI(4)P marked by PH domain of FAPP (RFP::PH-FAPP) associates with Golgi apparatus in *Drosophila* tissues (*Polevoy et al., 2009*). To probe the plasma membrane pool of PI(4)P more specifically, we took advantage of a PI(4)P-GFP reporter 2xOshPH::GFP [PI(4)P-GFP], the GFP-tagged PH domain of yeast Osh2, which is known to detect multiple pools of PI(4)P in yeast and mammalian fibroblasts (*Roy and Levine, 2004*). Strikingly, when it was expressed under *insc*-Gal4 driver in neuroblasts, the intense signal of PI(4)P reporter was uniformly cortical from interphase to anaphase (*Figure 5B* and *Figure 5—figure supplement 1B*). Interestingly, PI(4)P appeared to be enriched at cleavage furrow in telophase neuroblasts (*Figure 5B–D*, 73.1%, n = 78). These observations suggest that PI(4)P is localized to the plasma membrane in neuroblasts.

Next, we investigated whether the localization of PI(4)P in neuroblasts depended on Vib function. To this end, PI4P-GFP reporter was introduced into *vib*$^{133/j5A6}$ trans-heterozygous larval brains. Strikingly, PI(4)P-GFP in 44% of interphase *vib*$^{133/j5A6}$ neuroblasts localized to the cytoplasm and forming large aggregates (*Figure 5B,C*, n = 157). Cytoplasmic Mira aggregation was also observed in these neuroblasts (*Figure 5B*, 57.3%, n = 129). At metaphase, PI(4)P-GFP reporter in *vib*$^{133/j5A6}$ was largely unaffected, likely due to its high abundance at this stage (data not shown). However, the enrichment of PI(4)P at the cleavage furrow was lost in 51.9% of telophase neuroblasts (*Figure 5B,C*, n = 54). This observation is supported by the measurement of PI(4)P-GFP pixel intensity. Normalized against

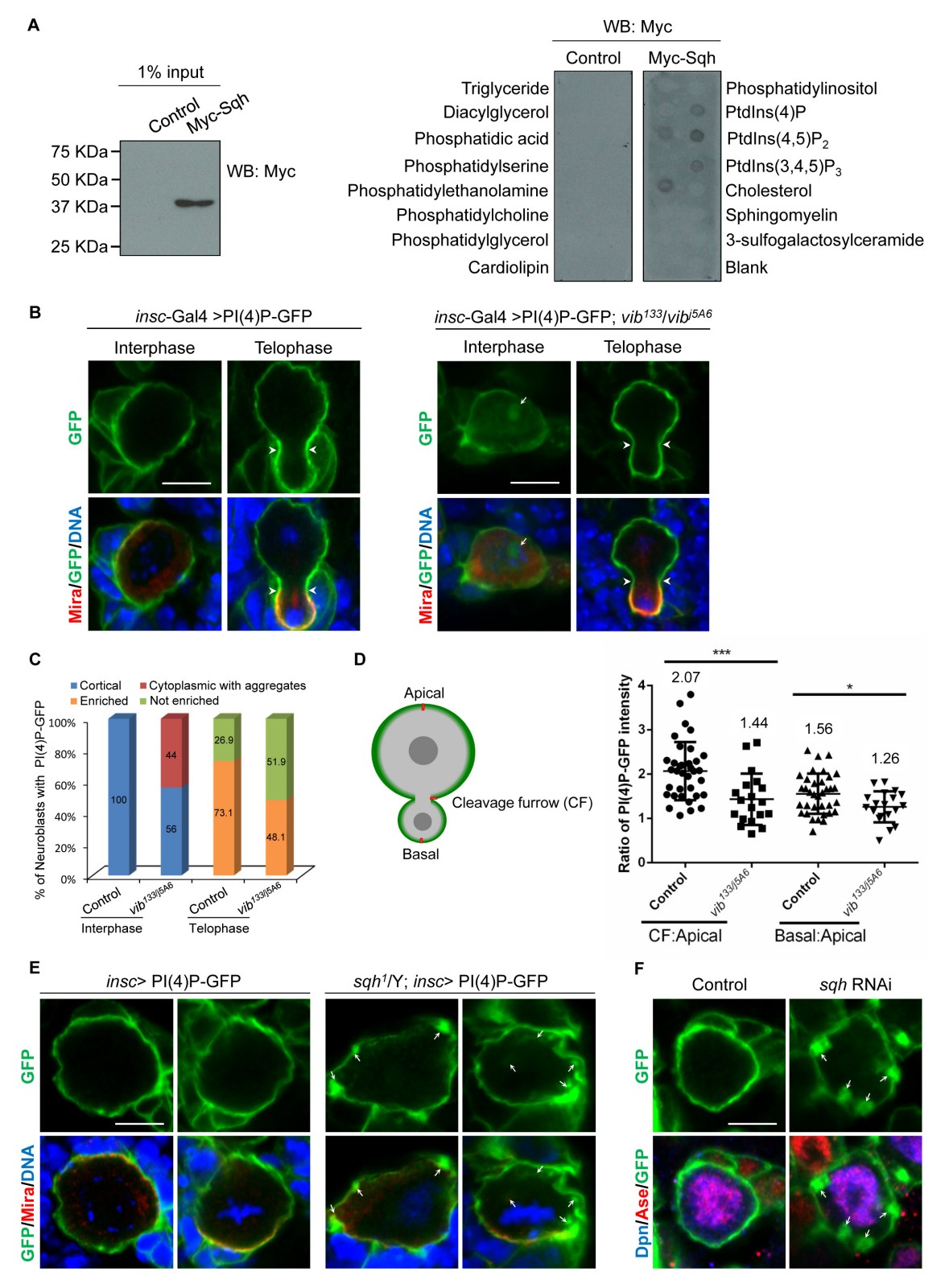

**Figure 5.** Sqh binds to phosphoinositide PI(4)P and localizes it to cell cortex in neuroblasts. (**A**) Control (empty Myc vector) and Myc-Sqh lysates from S2 cells were used for lipid-binding assay with lipid strips. Lipid strips were blotted with Myc antibody. 1% of lysate input used for lipid-binding assay is shown in the left panel. (**B**) Neuroblasts of PI(4)P-GFP (2xOsh2PH::GFP, PI(4)P reporter) driven by *insc*-Gal4 were labeled with GFP (green), Mira (red) and DNA (blue). Neuroblasts of *insc*-Gal4 driven PI(4)P reporter in *vib^133^/vib^i5A6^* were labeled with GFP (green), Mira (red) and DNA (blue). (**C**)
*Figure 5 continued on next page*

*Figure 5 continued*

Quantification of PI(4)P reporter localization in neuroblasts for (**B**). 'Enriched' in quantification legend refers to enriched localization at cleavage furrow, while 'not enriched' indicates no enrichment at cleavage furrow. (**D**) A schematic representation of measurement of the PI(4)P-GFP intensity for telophase neuroblasts shown in B. The red line shows the site where the measurement was taken in neuroblasts. The graph (to the right) shows the ratio of PI(4)P-GFP intensity at cleavage furrow (CF) or basal cortex to apical cortex in control and *vib*$^{133}$/*vib*$^{j5A6}$ expressing PI(4)P-GFP. (**E**) Neuroblasts of control and *sqh*$^1$/Y hemizygotes expressing PI(4)P-GFP driven by *insc*-Gal4 were labeled with Mira (red), GFP (green) and DNA (blue). (**F**) Neuroblasts of control (PI(4)P-GFP driven by *insc*-Gal4) and *sqh* knockdown with PI(4)P-GFP expression driven by *insc*-Gal4 were labelled with Dpn (blue), Ase (red) and GFP (green). Arrowheads indicate position of cleavage furrow and arrows indicate PI(4)P-GFP aggregates. Scale bars, 5 μm.
DOI: https://doi.org/10.7554/eLife.33555.014

The following figure supplement is available for figure 5:

**Figure supplement 1.** PI(4)P is localized to cell cortex in neuroblasts.
DOI: https://doi.org/10.7554/eLife.33555.015

the intensity at the apical cortex, wild-type neuroblasts showed an average of 2.07-fold enrichment (n = 34) of PI(4)P-GFP at the cleavage furrow, while in *vib*$^{133/j5A6}$ telophase neuroblasts, enrichment of PI(4)P-GFP decreased to 1.44-fold (n = 18) at cleavage furrow (*Figure 5D*). There was also a decrease of basal enrichment of PI(4)P-GFP at the basal cortex of *vib*$^{133/j5A6}$ neuroblasts, from 52.9% (average of 1.56 fold, n = 34) in wild-type neuroblasts to 33.3% (average of 1.26 fold, n = 18) in *vib*$^{133/j5A6}$ neuroblasts (*Figure 5D*).

Since we observed that Vib is required for the localization of both Sqh and PI(4)P on the neuroblast cortex and that Sqh binds to PI(4)P in vitro, we examined the role of Sqh in regulating PI(4)P localization in neuroblasts. We analyzed the localization of PI(4)P-GFP in *sqh*$^1$ hemizygotes, a hypomorphic allele of Sqh that survived to the third instar larval stage. Remarkably, 23.3% of interphase neuroblasts (n = 313) and 36.8% of metaphase neuroblasts (n = 19) displayed distinct PI(4)P-GFP aggregates in the cytoplasm (*Figure 5E*). Given the known role of Sqh in regulating cytokinesis, we could not analyze the localization of PI(4)P-GFP in telophase *sqh*$^1$ neuroblasts. Next, we knocked down *sqh* using *insc*-Gal4 and examined the localization of PI(4)P-GFP reporter in neuroblasts. While PI(4)P-GFP reporter was uniformly cortical in interphase control neuroblasts (*Figure 5F*, n = 105), *sqh* knockdown resulted in the formation of many distinct PI(4)P-GFP aggregates in interphase neuroblasts (*Figure 5F*, 30%, n = 40). We conclude that Sqh facilitates the membrane localization of PI(4)P in neuroblasts.

## Vib interacts with Sqh in vitro and in vivo in BiFC assays

Given that Vib is necessary for Sqh localization to the cell cortex, with both proteins mediating PI(4)P membrane localization in neuroblasts, we explored whether Vib physically interacts with Sqh. Due to Vib's robust PI/PC transfer activity, it is likely that interaction of Vib with its interacting proteins will be transient. Therefore, we adopted bimolecular fluorescence complementation (BiFC) assay (*Gohl et al., 2010*), a widely used method to probe protein-protein interactions that are transient or weak, due to the irreversibility of the BiFC complexes (*Shyu et al., 2008*). We generated chimeric proteins, Vib-Myc-N termini of YFP (Vib-Myc-NYFP), and C-termini of YFP (CYFP)-HA-Sqh constructs. Expression of these constructs with their respective negative controls, comprising of the matching half-YFP (refer to the legend), in S2 cells by *actin*-Gal4 did not result in any fluorescence signals (*Figure 6—figure supplement 1*). Remarkably, we observed intense YFP signals upon co-transfection of both constructs in S2 cells with *actin*-Gal4 (*Figure 6—figure supplement 1*). On the contrary, we did not observe YFP signals in S2 cells that co-express either Vib-Myc-NYFP with an unrelated control WAVE-HA-CYFP (*Gohl et al., 2010*) or CYFP-HA-Sqh with WAVE-Myc-NYFP (*Gohl et al., 2010*) (data not shown). As an additional control, Vib did not interact with Mira or Baz in BiFC assays (data not shown). This observation suggests that Vib and Sqh were in sufficiently close proximity to each other to allow the two-halves of YFP to merge in trans and reconstitute a functional fluorescence protein. To validate this interaction, we generated two truncated Vib proteins, Vib N-terminus (VibN) and Vib C terminus (VibC), and tested either truncated form of Vib abolished the interaction with Sqh in BiFC assays. Indeed, co-expression of either VibN-Myc-NYFP or VibC-Myc-NYFP with CYFP-HA-Sqh, driven by *actin*-Gal4 in S2 cells, did not result in any YFP fluorescence (*Figure 6—figure supplement 1*). As expected, neither of the negative control for these truncated proteins displayed YFP in the same BiFC assay (*Figure 6—figure supplement 1*).

Next, to examine if Vib and Sqh interact in vivo, we generated transgenes expressing NYFP-Myc-Vib and CYFP-HA-Sqh. Co-expression of NYFP-Myc with CYFP-HA-Sqh (metaphase, n = 44; telophase, n = 8) and NYFP-Myc-Vib with CYFP-HA (metaphase, n = 32; telophase, n = 10) under the neuroblast driver, *insc*-Gal4, did not result in YFP fluorescence (*Figure 6A*). By contrast, co-expression of both NYFP-Myc-Vib and CYFP-HA-Sqh resulted in YFP fluorescence in neuroblasts (*Figure 6A*). Despite NYFP-Myc-Vib being observed in the cytoplasm and CYFP-HA-Sqh at both cell cortex and spindle envelope in neuroblasts, YFP was found in the cytoplasm during interphase, then localized to the cortex as early as prophase and concentrated at the cleavage furrow at telophase (*Figure 6A*; metaphase, n = 44; telophase, n = 15). These data suggest that Vib interacts with Sqh at the cell cortex and cleavage furrow in neuroblasts.

To further validate the physical association between Sqh and Vib, we performed a proximity ligation assay (PLA), which allows detection of protein-protein interaction with high specificity and sensitivity in the form of fluorescence signal in situ (*Fredriksson et al., 2002*). We co-stained S2 cells with anti-Flag and anti-Myc antibodies to visualize the co-expression of proteins tagged with Flag or Myc. Quantifications of PLA foci were carried out only in cells with co-expression of Flag and Myc-tagged proteins (*Figure 6C,D*). In S2 cells expressing Flag and Myc controls, we did not observe any PLA fluorescence signal (*Figure 6C,D*; n = 96). In S2 cells that were co-transfected with Flag-Vib and Myc controls, 76.6% cells (n = 188) had no PLA signal and 19.2% cells displayed weak PLA fluorescence signal (≤10 foci), with the remaining 3.7% and 0.5% cells displaying PLA foci between 11 and 30 and >30, respectively. In control expressing Myc-Sqh with Flag, 86.6% cells (*Figure 6C,D*, n = 149) had no PLA signal and 13.4% of cells with weak PLA signal (≤10 foci). By contrast, 82.9% (*Figure 6C,D*, n = 216) of S2 cells expressing both Flag-Vib and Myc-Sqh showed PLA fluorescence: 15.3% of cells displayed strong PLA signal (>30 foci), 24.5% of cells showed moderate PLA signal (11–30 foci) and 43.1% of cells showed weak PLA signal (<10 foci). On average, controls Flag-Vib with Myc and Myc-Sqh with Flag resulted in 1.7 and 0.3 PLA foci per cell, respectively. By contrast, co-expressing Flag-Vib and Myc-Sqh resulted in 15.2 PLA foci per cell. These data reinforce that Vib and Sqh physically interact.

## PI4KIIIα, but not PI4KIIIβ or PI4KIIα, is required for asymmetric division and homeostasis of neuroblasts

There are three PI4-kinases in *Drosophila*, namely PI4KIIIα, PI4KIIIβ and PI4KIIα. PI4KIIIα is required for actin organization and cell polarity during oogenesis (*Tan et al., 2014*). PI4KIIIβ, also named Four wheel drive (Fwd), regulates cytokinesis of spermatocytes by localizing PI(4)P and Rab11 to the plasma membrane (*Polevoy et al., 2009*). PI4KIIα regulates membrane trafficking during the formation of secretory granules in the salivary gland (*Burgess et al., 2012*). The role of these PI4Ks during neuroblast asymmetric division was previously unknown. Toward this end, we examined neuroblast homeostasis and asymmetric division in mutants or RNAi lines for these three PI4Ks. Loss of function of *PI4KII* did not perturb neuroblast homeostasis or cortical polarity of aPKC and Mira (*Figure 7—figure supplement 1A,B*). Similarly, neuroblast homeostasis and cortical polarity were unaffected upon loss of *PI4KIIIβ/Fwd* (*Figure 7—figure supplement 1C* and data not shown).

*PI4KIIIα$^C$* is an EMS-induced mutant that carries a nonsense mutation at Trp855, leading to the generation of a truncated protein with its C-terminal half including the kinase domain deleted (*Yamamoto et al., 2014*). We observed ectopic neuroblasts in *PI4KIIIα$^C$* MARCM neuroblast clones, as evident by ectopic Dpn$^+$Ase$^+$ type I neuroblasts (*Figure 7A*, 44.8%, n = 29) and Dpn$^+$Ase$^-$ type II neuroblasts (*Figure 7A*, 30%, n = 20). Furthermore, 35.5% (n = 76) of *PI4KIIIα$^C$* MARCM clones were devoid of neuroblasts (data not shown), suggesting that neuroblast homeostasis was disrupted. Neuroblast homeostasis defects of *PI4KIIIα$^C$* clones were fully rescued by a PI4KIIIα genomic rescue construct (*Figure 7—figure supplement 1D*, type I clones, n = 81; type II clones, n = 36).

We next investigated whether PI4KIIIα is required for asymmetric division of neuroblasts. Depletion of *PI4KIIIα* appears to result in a delay of mitotic entry, as very few mitotic neuroblasts were found in MARCM clones generated. Apical proteins aPKC (42%, n = 12) and Par-6 (50%; n = 20) delocalized to the cytoplasm (*Figure 7B,C*). The delocalization of aPKC into cytoplasm presumably causes depletion of neuroblasts in *PI4KIIIα* mutants. Likewise, basal protein Mira either delocalized to the cytoplasm or mis-localized to the mitotic spindle in in *PI4KIIIα* mutants (*Figure 7B,C*, 77%, n = 13). Similarly, Numb crescent became punctate or delocalized in metaphase neuroblasts of *PI4KIIIα$^C$* (*Figure 7B,C*, 40%; n = 20). The delocalization of basal proteins appears to cause ectopic

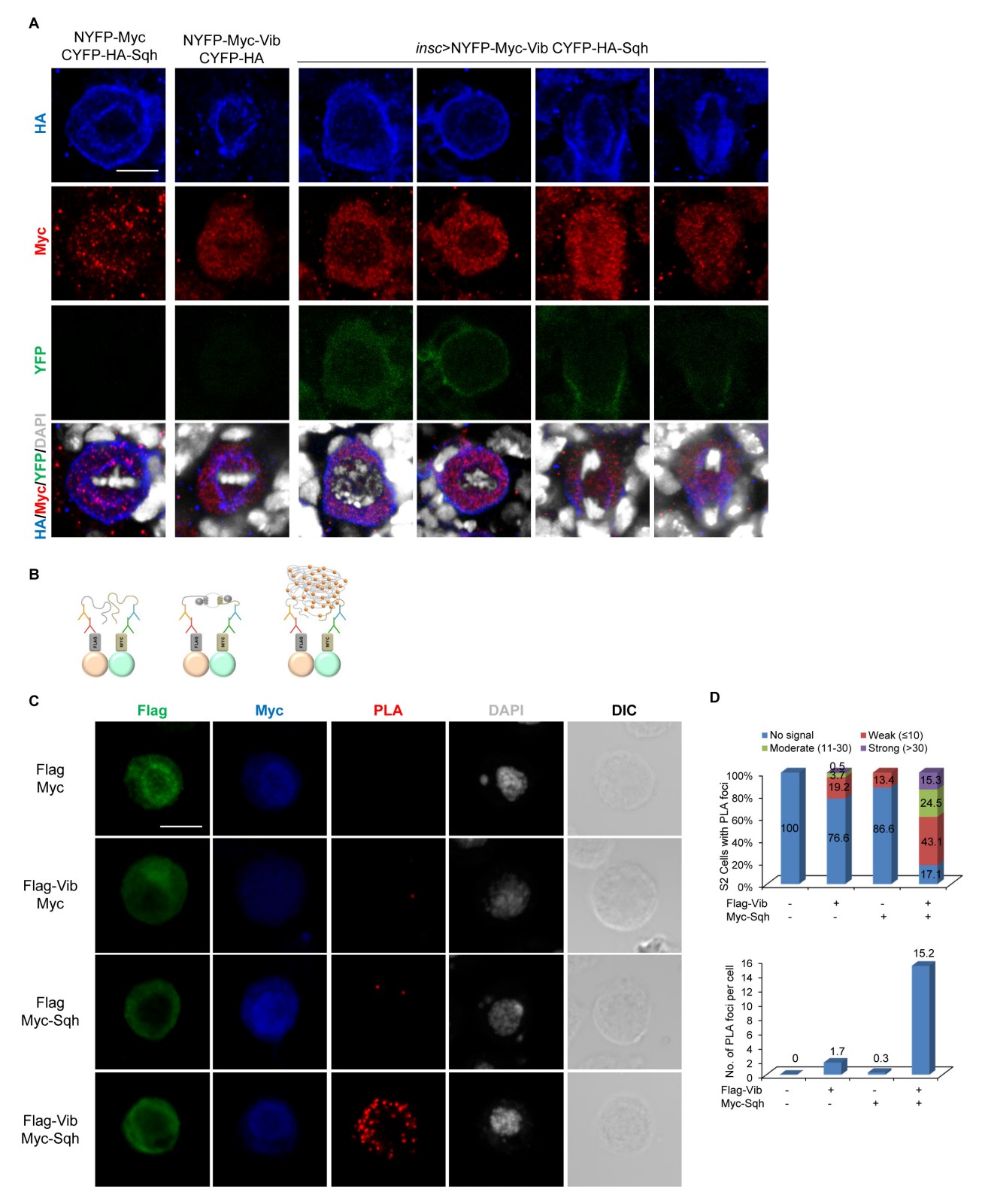

**Figure 6.** Vib interacts with Sqh in larval brain neuroblasts. (**A**) In vivo BiFC assay between Vib and Sqh. NYFP-Myc-Vib and CYFP-HA-Sqh were co-expressed in neuroblasts by *insc*-Gal4, stained with HA (red), Myc (red) and DAPI (grey), and detected for YFP fluorescence (green). Controls were NYFP-Myc-Vib with CYFP-HA control and CYFP-HA-Sqh with NYFP-Myc Control. Scale bars, 5 μm. (**B**) Schematic representation of proximity ligation assay performed on S2 cells (refer to Materials and methods). (**C**) In situ PLA assay between Flag-Vib and Myc-Sqh in S2 cells. S2 cells transfected with

*Figure 6 continued on next page*

*Figure 6 continued*
the indicated plasmids were stained with Flag (green), Myc (blue) and DAPI (grey) and detected for PLA signal (red). Cell outline was shown by differential interference contrast (DIC) images. Scale bar, 5 μm. (**D**) Graph showing the percentage of S2 cells that expressed no PLA signal, weak (≤10 foci), moderate (11–30 foci) and strong (>30 foci) PLA signals for (**C**). Quantification for the average number of PLA foci per cell in (**C**).
DOI: https://doi.org/10.7554/eLife.33555.016
The following figure supplement is available for figure 6:

**Figure supplement 1.** Vib interacts with Sqh in Bimolecular fluorescence complementation.
DOI: https://doi.org/10.7554/eLife.33555.017

neuroblasts observed in *PI4KIIIα* mutants. These observations suggest a role of PI4KIIIα is required for asymmetric division and homeostasis of neuroblasts.

## PI4KIIIα localizes Sqh to the cell cortex in neuroblasts

To determine whether Sqh was a common target of Vib and PI4KIIIα during asymmetric division of neuroblasts, we sought to investigate whether Sqh localization is dependent on PI4KIIIα in neuroblasts. To this end, we examined the localization of Sqh::mCherry in *PI4KIIIα^C* MARCM clones. Since Sqh::mCherry is mostly cytoplasmic during interphase, we focused our analysis on mitotic neuroblasts. In control MARCM clones, 46% of metaphase neuroblasts showed strong cortical localization of Sqh::mCherry, 23% was weakly cortical and 31% was cytoplasmic (*Figure 7D*, n = 11). However, a vast majority of metaphase neuroblasts from *PI4KIIIα^C* MARCM clones displayed either cytoplasmic (36.4%) or weakly cortical localization (*Figure 7D*, 54.5%, n = 11). This observation suggests that similar to Vib, PI4KIIIα is important for anchoring Sqh to the cell cortex in dividing neuroblasts. Consistently, Sqh1P cortical localization also depended on PI4KIIIα in neuroblasts (*Figure 7—figure supplement 1E*). In the controls, while 12% were weakly cortical and the remaining 20% were cytoplasmic, 68% of metaphase neuroblasts showed cortical Sqh1P localization (*Figure 7—figure supplement 1E*, n = 25). By contrast, only 5.6% (n = 18) of metaphase neuroblasts of PI4KIIIα^C showed obvious cortical localization and the rest of them were either cytoplasmic (44.4%, n = 18) or weakly cortical (*Figure 7—figure supplements 1E*, 50%, n = 18). At telophase, Sqh1P localization at the cleavage furrow was only observed in 44.4% of PI4KIIIα^C neuroblasts (*Figure 7—figure supplement 1E*, n = 9), while majority of control telophase neuroblasts showed Sqh1P localization at the cleavage furrow (*Figure 7—figure supplements 1E*, 84.2%, n = 19). These observations strongly suggest that PI4KIIIα plays an important role in localizing Sqh to the cell cortex in neuroblasts.

Next, we assessed the effect of pharmacological inhibition of PI4KIIIα on asymmetric division of neuroblasts by phenylarsine oxide (PAO), an inhibitor of PI4-kinases at a low concentration of 2.5 μM (*Bryant et al., 2015*). In the mock treatment (DMSO) to wild-type larval brains, 82.4% of metaphase Sqh::mCherry were cortically localized and only 17.6% of neuroblasts showed cytoplasmic distribution (*Figure 7E*, n = 51). By contrast, with PAO treatment, Sqh::mCherry became cytoplasmic in 59% (*Figure 7E*, n = 117) of metaphase neuroblasts. Likewise, compared with basal Mira crescent (100%, n = 48) in the mock treatment, 25% (n = 64) of neuroblasts upon PAO treatment showed cytoplasmic Mira localization at metaphase (*Figure 7F*), which was consistent with above observations in PI4KIIIα^C neuroblasts. However, Rok-GFP localization was unaffected following PAO treatment. Similar to mock (DMSO treatment), 92% (n = 112) of metaphase neuroblasts with cortical Rok-GFP localization and 76.5% (n = 17) of telophase with Rok-GFP enriched at the cleavage furrow were observed (*Figure 7—figure supplement 2*). This result suggests that Rok localization is likely independent of PI4KIIIα. Taken together, these observations reinforce the role of PI4KIIIα in Sqh localization to the cell cortex as well as asymmetric division.

## Discussion

Here, we show a new role for *Drosophila* PITP, Vibrator/Giotto in asymmetric cell division and homeostasis of neuroblasts by interacting and anchoring non-muscle myosin II light chain, Sqh. Importantly, lipid binding/transfer activities, particularly PI binding and transfer activities of Vib, are critical for asymmetric division. We also show that Sqh binds to phosphoinositides including PI(4)P and that a pool of PI(4)P is localized at the cell cortex of neuroblasts in a Vib and Sqh-dependent manner. Finally, a PI4-kinase PI4KIIIα, an essential kinase for PI(4)P synthesis, is required for Sqh

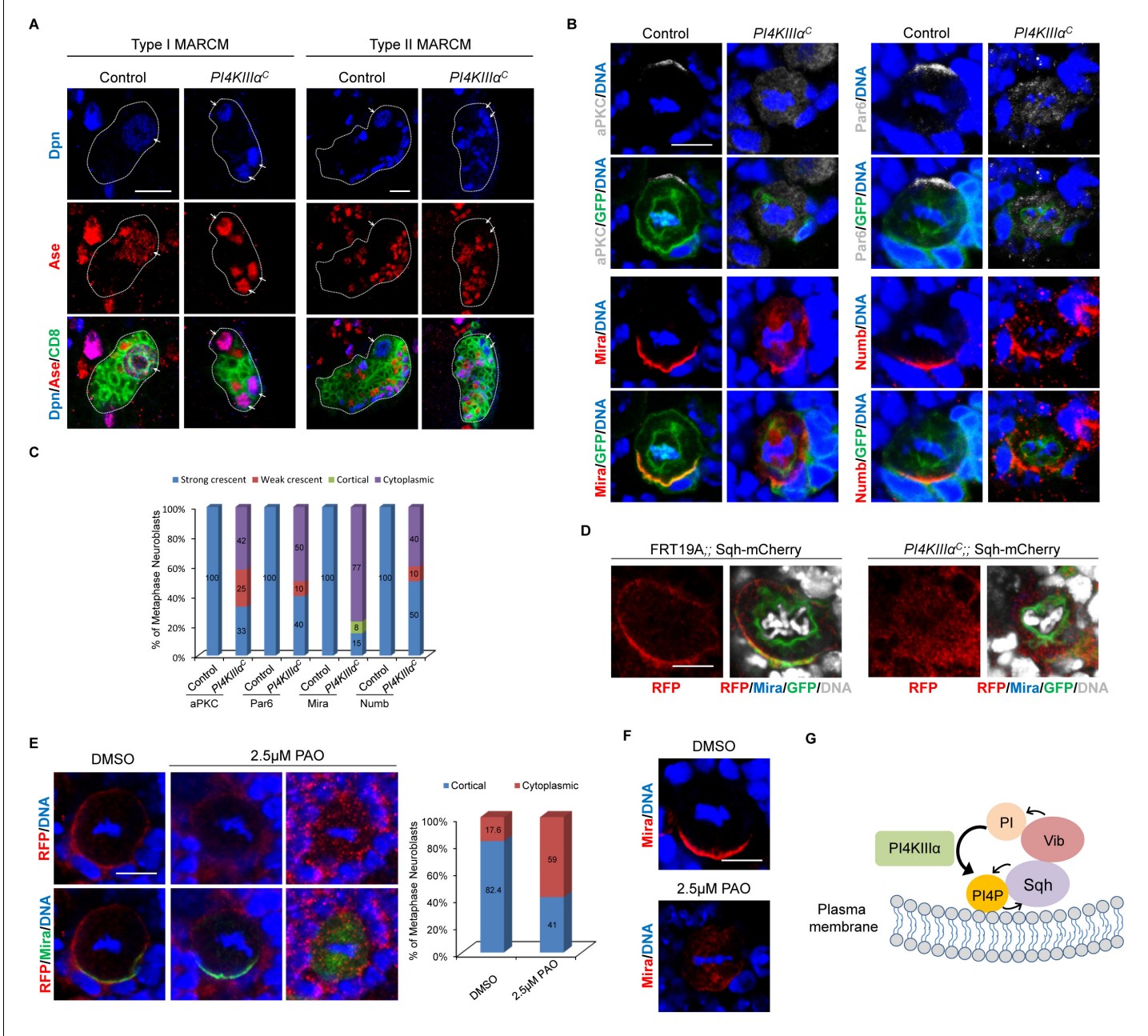

**Figure 7.** Loss of *PI4KIIIα* perturbs neuroblast homeostasis and asymmetric cell division of neuroblasts. (**A**) MARCM clones of control (FRT19A) and *PI4KIIIα^C* were labeled with Dpn (blue), Ase (red) and CD8 (green). Arrows indicate neuroblasts. Clones were marked by CD8::GFP and indicated by white dotted line. (**B**) Neuroblasts of control and *PI4KIIIα^C* MARCM clones were labeled with aPKC or Par-6 (grey), Mira or Numb (red), GFP (green) and DNA (blue). (**C**) Quantification for (**B**). (**D**) Neuroblasts of control (FRT19A;; Sqh::mCherry) and *PI4KIIIα^C*;; Sqh::mCherry MARCM clones were labeled with RFP (red), Mira (blue), GFP (green) and DAPI (grey). (**E**) Neuroblasts of Sqh::mCherry treated with mock (DMSO) or 2.5 μM PAO in DMSO were stained with RFP (red), Mira (green) and DNA. Quantification was shown next to images. (**F**) Neuroblasts treated with mock or 2.5 μM PAO in DMSO were stained with Mira (red) and DNA (blue). (**G**) A working model. Scale bars, 10 μm (**A**); 5 μm (**B, D–F**).
DOI: https://doi.org/10.7554/eLife.33555.018

The following figure supplements are available for figure 7:

**Figure supplement 1.** PI4KIIIα, but not PI4KIIIβ or PI4KIIα regulates neuroblast asymmetric division and homeostasis.
DOI: https://doi.org/10.7554/eLife.33555.019

**Figure supplement 2.** PI4KIIIα does not regulate Rok localization in neuroblasts.
DOI: https://doi.org/10.7554/eLife.33555.020

localization and asymmetric division, similar to Vib. Given the lipid-binding and transfer function of Vib, it is conceivable that it anchors Sqh to the cell cortex prior to the asymmetric localization of Sqh and basal proteins in neuroblasts, rather than directly regulating asymmetric localization of Sqh or basal proteins. In addition, Vib's localization in neuroblast cortex lacks apparent asymmetry, while it is required for the asymmetric localization of apical proteins aPKC and Par-6, but not Baz. We speculate that Vib might also play a role in membrane localization of aPKC or Par-6 in neuroblasts. We also show that Vib likely interacts with Sqh to regulate asymmetric division through its stimulation of PI4KIIIα to produce a plasma membrane pool of PI(4)P that, in turn, specifies cortical targeting of Sqh in neuroblasts. This reciprocal regulation between Myosin and lipid transfer/synthesis may be important for asymmetric division of neuroblasts.

The asymmetric division defects of *vib*-depleted neuroblasts were not a consequence of cytokinesis failure, because the VibT63E mutation, which rendered Vib deficient in PI binding/transfer with reduced PC binding/transfer, was able to rescue the defects of cytokinesis, but not neuroblast polarity, observed in *vib*-depleted neuroblasts. This finding suggests that PC binding/transfer function of Vib may be more important for cytokinesis, while PI binding/transfer may play a more prominent role in neuroblast polarity. Therefore, the pleotropic functions of Vib may be separable by cooperating with different downstream proteins during cytokinesis and asymmetric division of neuroblasts. Supporting this notion, depletion of PI4KIIIα resulted in loss of neuroblast polarity without blocking cytokinesis. Thus, it is conceivable that Vib and PI4KIIIα-dependent cortical localization of Sqh regulates asymmetric division, but not cytokinesis.

PI(4)P is a functionally diverse inositol lipid in cells (*Gassama-Diagne and Payrastre, 2009*; *Hammond et al., 2009*). Apart from being the major precursor of PI(4,5)P2 biosynthesis, PI(4)P also plays numerous crucial functions such as regulation of protein trafficking at the level of the Golgi complex, and sphingolipid and sterol biosynthetic trafficking (*Matsuoka et al., 1998*; *Blumental-Perry et al., 2006*; *Olkkonen and Li, 2013*). We show that PI(4)P can be detected on the cell cortex of neuroblasts with cleavage furrow enrichment at telophase. Moreover, we show that the distribution of PI(4)P on the plasma membrane is mediated by both Vib and Sqh in neuroblasts. Supporting this notion, Sqh binds to PI(4)P in vitro. Although the apical protein Baz, an apical protein, is known to be targeted to the plasma membrane via direct interactions with PI(3,4,5)P3 and PI(4,5)P2 (*Heo et al., 2006*; *Krahn et al., 2010*). Interestingly, the role of Vib in neuroblast asymmetric division that we report here is substantially independent of Baz function.

Mammalian PI4-kinases are responsible for the generation of PI(4)P within cells and exhibit different subcellular distribution (*Clayton et al., 2013*). While type II PI4-kinases and type III PI4KIIIβ localize to intracellular membranes including trans-Golgi network (TGN), endoplasmic reticulum (ER), endosomal and trafficking vesicles, PI4KIIIα localizes to the plasma membrane, early cis-Golgi compartments and nucleolus (*D'Angelo et al., 2008*). The known distribution of various PI4 kinases and the fact that PI4KIIIα is the major kinase responsible for plasma membrane PI(4)P biosynthesis (*Roy and Levine, 2004*; *Balla et al., 2005*) well support the unique role of PI4KIIIα among three *Drosophila* PI4-kinases in neuroblast asymmetry. *Drosophila* PI4kIIIα is also required for plasma membrane integrity and oocyte chamber polarity potentially through regulating actin organization (*Tan et al., 2014*). Interestingly, we found that actin localization labeled by Phalloidin was unaffected in both *vib* and *PI4KIIIα* mutant neuroblasts (data not shown), suggesting that both Vib and PI4KIIIα likely regulate neuroblast polarity independent of actin organization. Mammalian PI4-Kinases are associated with cancer and neurological diseases including schizophrenia (*Furuta et al., 2003*; *Jungerius et al., 2008*; *Clayton et al., 2013*). Thus, our findings on the role of *Drosophila* PI4KIIIα in neuroblast asymmetric division may shed light on the mechanisms underlying these diseases.

*Drosophila* non-muscle myosin II RLC Sqh regulates asymmetric localization of basal cell fate determinants in neuroblasts (*Barros et al., 2003*). Similarly, *C. elegans* non-muscle myosin II exhibits polarized distribution and mediates asymmetric localization of Par-3 to the anterior cortex during the asymmetric division of the one-cell stage embryo (*Severson and Bowerman, 2003*; *Ou et al., 2010*). Here, we show that Vib is critical for anchoring Sqh to the neuroblast cortex. The asymmetric localization of Sqh in neuroblasts has been shown to be dependent on apical protein Pins, but not aPKC, Baz or canonical centralspindlin pathway including Pavarotti (Pav) (*Gatt and Glover, 2006*). The novel role of Vib in anchoring Sqh appears to be independent of Pins, since Vib is not important for Pins cortical polarity in neuroblasts. Together with our finding on the role of PI4KIIIα in Sqh

localization, we demonstrate that the Vib and PI4KIIIα-dependent inositol lipid PI(4)P is important for Sqh cortical localization in neuroblasts.

Myosin is known to associate with lipids indirectly through F-actin and the direct link between myosin and lipids was unclear. Our data on the association between Vib and Sqh provides a novel link between PITP-dependent PI(4)P pools and myosin localization in dividing cells. In addition, we show the binding of Sqh to phospholipids including PI(4)P, directly linking myosin to phospholipids. Very recently, mammalian non-muscle myosin II was shown to bind to liposomes containing multiple acidic PIPs through its RLC-binding site, but not to liposomes that contain only PC (*Liu et al., 2016*). Our finding that Sqh mediates membrane localization of PI(4)P suggests that Sqh may reinforce the cortical localization of itself through an interaction with Vib and PI(4)P in neuroblasts. Whether this reciprocal regulation between Myosin and lipid synthesis is similarly employed in other cell types will be of great interest in future studies.

## Materials and methods

### Fly stocks and genetics

Fly stocks and genetic crosses were raised at 25°C unless otherwise stated. Fly stocks were kept in vials or bottles containing standard fly food (0.8% *Drosophila* agar, 5.8% Cornmeal, 5.1% Dextrose and 2.4% Brewer's yeast). The following fly strains were used: $vib^{133}$, $vib^{1105}$, UAS-NYFP-Myc, UAS-CYFP-HA, UAS-NYFP-Myc-Vib, UAS-CYFP-HA-Sqh, UAS-Vib::Venus, UAS-VibT63A::Venus, UAS-VibT63E::Venus, UAS-PITPα::Venus and UAS-PITPβ::Venus (this study), $fwd^3$, P{w+, PI4KIIIα} FRT80B and $PI4KII\Delta$ (Julie, A. Brill), $sqh^{Ax3}$; p[w;sqh-GFP42], UAS-Cdc42::GFP (Akira Chiba), $baz$, FRT19A, w, Ubi-Rok-GFP (I) and w;; Ubi-Rok-GFP (III) (Bellaiche, Y), UAS-WAVE-HA-CYFP and UAS-Abi-Myc-NYFP(Bogdan, S). The following stocks were obtained from Bloomington *Drosophila* Stock Center: insc-Gal4 (BDSC#8751), elav-Gal4 (BDSC#8765), P{lacW}$vib^{j5A6}$/TM6B,Tb (BDSC#12144), P{lacW}$vib^{j7A3}$/TM3, Sb (BDSC#10308), Vib deficiency w; Df(3R)BSC850/TM6C,Sb,cu (BDSC#27922), $sqh^{AX3}$, FRT19A/FM7c (BDSC#25712), sqh RNAi (BDSC#33892), w; Sqh-mCherry (BDSC#59024), $PI4KIIIα^C$, FRT19A/FM7c (BDSC#57112), w; UAS-2xOsh2PH-GFP (BDSC#57353), w; UAS-2xOsh2PH-GFP/CyO; Pri/TM6B, Tb (BSDC#57352).

Crosses for RNAi knockdown and overexpression were incubated at 25°C for 24 hr before transferring to 29°C, where the larvae were aged for 72 hr. Wandering third instar larvae were dissected and processed for immunohistochemistry staining.

### Immunohistochemistry and immunoblotting

Immunohistochemistry on third instar larval brains was performed as previously described (*Wang et al., 2006*). The primary antibodies used were guinea pig anti-Dpn (1:1000), rabbit anti-Ase (1:500), rat anti-CD8 (1:200; Invitrogen), rabbit anti-aPKCζ C20 (1:100; Santa Cruz Biotechnology, Inc.), rabbit anti-PKC iota phospho-T555 + T563 (1:100; abcam), rabbit anti-Insc (1:1,000), guinea pig anti-Gαi (1:200; F. Yu) , rat anti-Brat (1:100; R.P. Wharton), mouse anti-α-tubulin (1:200; Sigma-Aldrich), mouse anti-Mira (1:40; F. Matsuzaki), guinea pig anti-Baz (1:500; A. Wodarz), guinea pig anti-Numb (1:100), rabbit anti-PON (1:100; Y.N. Jan), rabbit anti-GFP (1:1,000; Molecular Probes), rabbit anti-RFP (1:100; Abcam), rabbit anti-Sqh (1:250; Adam C. Martin), guinea pig anti-Sqh1P (1:500, Robert E. Ward IV), mouse anti-Myc (1:200, Abcam) and rabbit anti-HA (1:100, Sigma-Aldrich). The secondary antibodies used were conjugated with Alexa Fluor 488, 555 or 647 (Jackson laboratory).

To extract protein from larval brains, 50 wandering third instar larvae were dissected and brains were homogenized using RIPA buffer 50 mM Tris-HCl, pH 7.5, 150 mM NaCl, 1 mM EDTA, 1% Triton X-100, 0.5% sodium deoxycholate, and 0.1% SDS). Protein lysate were subjected to SDS-PAGE and western blotting. Antibodies used for western blotting were rabbit anti-Vib (this study; 1:2,000), mouse anti-α-tubulin (1:10,000; Sigma-Aldrich), mouse anti-Flag (1:2,000; Sigma-Aldrich), mouse anti-Myc (1:2,000; abcam), donkey α-mouse IgG HRP (1:5000, Pierce, SA1-100) and donkey α-rabbit IgG HRP (1:5000, Pierce, SA1-200).

## Clonal analysis

MARCM clones were generated as previously described (*Lee and Luo, 1999*). Briefly, larvae were heat shocked at 37°C for 90 min at 24 hr ALH and at 10–16 hr after the first heat shock. Larvae were further aged for 3 days at 25°C, and larval brains were dissected and processed for immunohistochemistry. In our analysis on neuroblast homeostasis in clones, neuroblasts with obvious cytokinesis defects (polyploidy cells) were excluded. Neuroblasts with two (Dpn+) nuclei in the same cytoplasm were also excluded in our analysis. We only counted ectopic neuroblasts that were clearly separated cells based on CD8-GFP, which marked the outline of cells.

## Quantification of mitotic spindle orientation

Mitotic spindle orientation was quantified for metaphase neuroblasts labeled with Insc, α-tubulin and DNA (Topro3). Apicobasal axis was inferred by a line that is perpendicular to the Insc crescent whereas the mitotic spindle represents the spindle axis. Spindle orientation which is denoted by the angle between the apicobasal and spindle axis, was measured and quantified.

## Measurement of PI(4)P-GFP intensity

The pixel intensity of PI(4)P-GFP at the apical cortex, basal cortex and cleavage furrow of telophase neuroblasts were measured using ImageJ. Corresponding intensity was recorded and used to tabulate the ratio of PI(4)P-GFP intensity at cleavage furrow (CF) to PI(4)P-GFP intensity at apical cortex (CF:Apical) and the ratio of PI(4)P-GFP intensity at basal cortex to PI(4)P-GFP intensity at apical cortex (Basal:Apical), respectively.

## Phenylarsine oxide (PAO) treatment of larval brains

Wandering third instar larvae were dissected and the harvested brains were incubated in supplemented Schneider's medium (supplemented with 10% FBS, 50 U/ml penicillin, 50 µg/ml Streptomycin, 0.02 mg/ml insulin, 20 mM glutamine and 0.04 mg/ml glutathione) containing 2.5 µM PAO (Sigma-Aldrich) for 2 hr at RT. Mock treatment was done in supplemented Schneider medium added with DMSO. Larval brains were rinsed twice with PBS and processed for immunohistochemistry. For Ubi-Rok-GFP, larval brains were washed twice with PHEM buffer (60 mM PIPES, 25 mM HEPES, 10 mM EGTA and 4 mM MgSO$_4$), fixed with 4% formaldehyde in PHEM for 22 min and processed for immunohistochemistry.

## Molecular cloning

Expressed-sequence tags (EST) SD01527 (Vib), LD14743 (Sqh), and SD12145 (PI4KIIIα) were obtained from *Drosophila* Genomics Resource Center (DGRC). Full length coding sequence of Vib and Sqh were amplified by PCR and cloned into pENTR/D-TOPO vector (Invitrogen) or pDONR221 vector (Invitrogen) using BP Clonase II (Invitrogen) according to the manufacturer's protocol. Entry clone of Vib and Sqh were subsequently cloned into various destination vectors using LR clonase II (Invitrogen) according to manufacturer's protocol. To remove the RfB cassette from the pUAS-NYFP-Myc-RfB cassette and pUAS-CYFP-HA-RfB cassette vector, a short oligo was first cloned into pDONR221 using BP clonase II and subsequently cloned into pUAS-NYFP-Myc and pUAS-CYFP-HA using LR clonase II. Primers used in this study are listed in *Table 1*. The destination vectors used were pTW, pTWV, pTVW, pAFW and pAMW, which were obtained from DGRC. BiFC destination vectors, pUAST-NYFP-Myc, pUAST-CYFP-HA, pUAST-Myc-NYFP and pUAST-HA-CYFP were generous gifts from Bogdan, S.

## Generation of transgenic flies

UAS-Vib, UAS-Vib::Venus, UAS-VibT63A::Venus, UAS-VibT63E::Venus, UAS-PITPα::Venus, UAS-PITPβ::Venus, UAS-NYFP-Myc, UAS-CYFP-HA, UAS-NYFP-Myc-Vib and UAS-CYFP-HA-Sqh transgenic flies were generated by P-element-mediated transformation (BestGene Inc.). BDSC 8622 [yw; P{CaryP}attP2] was used as the injection stock for site-specific insertion of UAS-NYFP-Myc, UAS-CYFP-HA, UAS-NYFP-Myc-Vib and UAS-CYFP-HA-Sqh into chromosomal location 68A4 (BestGene Inc.).

**Table 1.** List of primers used

| Primer name | Primer sequence (5′—3′) | Purpose |
|---|---|---|
| Vib pENTR F | CACC ATG CAG ATC AAA GAA TTC CGT GTG | Generate Vib pENTR |
| Vib pENTR R | TTA ATC GGC ATC CGC GCG C | |
| Vib pENTR CT R | ATC GGC ATC CGC GCG CAT | Generate Vib pENTR without stop codon |
| Vib T63A F | TAC AAT TCC GGT CAA TAT GCC TAT AAG | Generate T63A mutation |
| Vib T63A R | CTT ATA GGC ATA TTG ACC GGA ATT GTA | |
| Vib T63E F | TAC AAT TCC GGT CAA TAT GAG TAT AAG | Generate T63E mutation |
| Vib T63E R | CTT ATA CTC ATA TTG ACC GGA ATT GTA | |
| Sqh pENTR F | CACC ATG TCA TCC CGT AAG ACC GC | Generate Sqh pENTR |
| Sqh pENTR R | TTA CTG CTC ATC CTT GTC CTT GG | |
| PITPα pENTR F | CACC ATG GTG CTG CTC AAG GAA TA | Generate PITPα pENTR |
| PITPα pENTR R | GTC ATC TGC TGT CAT TCC TTT CA | |
| PITPβ pENTR F | CACC ATG GTG CTG ATT AAG GAA TTC CG | Generate PITPβ pENTR |
| PITPβ pENTR R | GGC ATC AGC AGC CGA CGT GC | |
| Modifier pD221 F | GGGG ACAAGTTTGTACAAAAAAGCAGGCTTC GGTACCGCTGAAACGAAGTTAAACTTTGAGGTGTACGG GTAAGTATTAGAAAGCAGGACTAAACG | Modify pDONR221 to remove RfB cassette from BiFC constructs |
| Modifier pD221 R | GGGG ACCACTTTGTACAAGAAAGCTGGGTC AAGCTTCTATTCAATGGTCCGGCGGCCGACGACATGAGG ATATGGTCGTTTAGTCCTGCTTTCTAA | |

DOI: https://doi.org/10.7554/eLife.33555.021

## Generation of anti-Vib antibody

Full length of Vib tagged with N-terminal His was expressed in bacterial cells, purified and used to immunize rabbits to raise polyclonal antibody, followed by affinity purification (Genscript).

## Cell lines and transfection

*Drosophila* S2 cells (CVCL_Z232) originally from William Chia's laboratory (with a non-authenticated identity but have been used in the laboratory for the past 10 years) were cultured in Express Five serum-free medium (Gibco) supplemented with 2 mM Glutamine (Thermo Fisher Scientific). S2 cell culture used in this study is free of contamination of Mycoplasma, due to the absence of small speckles of DAPI staining outside of the cell nucleus. For transient expression of proteins, S2 cells were transfected using Effectene Transfection Reagent (QIAGEN) according to manufacturer's protocol. S2 cells were harvested 48 hr after transfection and were homogenized using lysis buffer (25 mM Tris pH8/27.5 mM NaCl/20 mM KCl/25 mM sucrose/10 mM EDTA/10 Mm EGTA/1 mM DTT/10% (v/v) glycerol/0.5% Nonidet P40) with Complete Proteases inhibitors (Roche) for 30 min at 4°C. Cell lysates were subjected to SDS-PAGE and western blotting.

## Bimolecular fluorescence complementation

In vitro bimolecular fluorescence complementation assay (*Gohl et al., 2010*) was performed using S2 cells. $1 \times 10^6$ cells were seeded onto Poly-L-lysine coated coverslips (Iwaki). S2 cells were transfected using Effectene Transfection Reagent (QIAGEN) with *act*-Gal4 and the BiFC constructs each at 0.2 µg per well, respectively. The BiFC constructs used were: pUAS-NYFP-Myc control and pUAS-CYFP-HA control, pUAS-NYFP-Myc-Vib, pUAS-Vib-Myc-NYFP, pUAS-VibN-Myc-NYFP, pUAS-VibC-Myc-NYFP, pUAS-CYFP-HA-Sqh, pUAS-NYFP-Myc and pUAS-CYFP-HA (this study), pUAS-WAVE-HA-CYFP, pUAS-WAVE-Myc-NYFP, pUAS-Abi-HA-CYFP, pUAS-Abi-Myc-NYFP and the pUAST-BiFC vectors (Bogdan, S.) as control.

48 hr after transfection, growth medium was removed and S2 cells were rinsed with cold PBS before fixing with 4% EM grade Formaldehyde in PBS for 15 min. Fixed S2 cells were rinsed three times with PBS-T (1xPBS + 0.1% Triton-X100) and blocked with 5% BSA in PBS-T for 1 hr before primary antibody incubation at RT for 2 hr. S2 cells were then rinsed three times with PBS-T and were

incubated with secondary antibodies in PBS-T for 1 hr at RT. Coverslips coated with immuno-stained S2 cells were mounted on to glass slides using vector shield (Vector Laboratory) for confocal microscopy. In vivo bimolecular fluorescence complementation was performed by expressing the BiFC vectors and constructs using *insc*-Gal4. Crosses were set up and incubated at 25°C for 24 hr before transferring to 29°C. Larvae were aged for 72 hr and wandering third instar larvae were dissected and processed for immunohistochemistry staining.

## Lipid binding and transfer assays

L-α-phosphatidylcholine, (chicken) egg PtdCho and L-α-phosphatidic acid (chicken) egg were purchased from Avanti Polar Lipids (Alabaster, AL). 1-palmitoyl-2-decapyrenyl-*sn*-glycero-3-phosphocholine, PyrPtdCho and 1-palmitoyl-2-decapyrenyl-*sn*-glycero-3-phosphoinositol, PyrPtdIns, were generous gifts from Dr. Pentti Somerharju (Helsinki University); synthesis as described (*Gupta et al., 1977*; *Somerharju and Wirtz, 1982*). 2,4,6-Trinitrophenylphosphatidyl-ethanolamine, TNP-PtdEtn was prepared from phosphor-ethanolamine as previously described (*Gordesky and Marinetti, 1973*) and purified by silica gel column chromatography. The concentration of all phospholipid solutions was determined as previously described (*Rouser et al., 1970*). PyrPtdCho and PyrPtdIns concentrations were determined spectroscopically using $\varepsilon = 42,000$.

The PyrPtdCho and PyrPtdIns binding and transfer measurements were essentially done as previously described (*Somerharju et al., 1987*). For the binding measurements donor vesicles were made from eggPtdCho, PyrPtdCho/PyrPtdIns and TNP-PtdEtn. Solvent was evaporated and the lipid film was re-suspended in 10 µl of EtOH. The solution was then injected into 2 mL of low phosphate buffer (25 mM Na2HPO4, 300 mM NaCl, pH 7.5) at 37°C and after 5–10 min incubation the solution was titrated with 0.1 nmol of indicated protein and fluorescence intensity was measured (Horiba Ltd. Kyoto, Japan). For transfer measurements, the donor vesicles were injected into low phosphate buffer containing acceptor vesicles. The resulting lipid film was then hydrated and sonicated on ice for 10 min. After donor vesicle addition, the solution was incubated for 5–10 min at 37°C. After which the fluorescence intensity was recorded as a function of time. To initiate protein mediated lipid transfer total 9 µg of protein was injected to achieve binding site saturation. The end-point assays measuring transport of [3 hr]-PtdIns were performed as previously described (*Bankaitis et al., 1990*; *Schaaf et al., 2008*).

## Protein-lipid-binding assay

The membrane lipid strips (P-6002, Echelon Biosciences) were blocked in 3% fatty acid free BSA (Sigma, A7030) in PBS-T [0.1% Tween 20 (T)] at 4°C overnight. S2 cells transfected with the indicated plasmids were lysed in lysis buffer (50 mM Tris-HCl, pH7.8, 150 mM NaCl and 0.5% Triton X100) for 30 min at 4°C. The strips were incubated with S2 cell lysate in 3% fatty-acid-free BSA in PBS-0.05% T for 1 hr at RT. Strips were washed in PBS-0.1% T for 10 min thrice, incubated with mouse anti-Myc (1:2,000; abcam) and subsequently proceeded similarly to western blot.

## Proximity ligation assay (PLA)

The rational of PLA is as follow: Secondary antibodies conjugated with PLA PLUS or PLA MINUS probe bind to anti-Flag and anti-Myc antibodies, respectively. During ligation, connector oligos hybridize to PLA probes and T4 ligases catalyze to form a circularized template. DNA polymerase amplifies circularized template, which is bound by fluorescently labeled complementary oligos allowing the interaction to be observed as a PLA foci within cells (Adopted from Duolink PLA, Merck). Proximity ligation assay was performed on S2 cells that were transfected with the indicated plasmids using Effectene Transfection Reagent (QIAGEN). The plasmids used were: control Myc, control Flag, Flag-Vib and Myc-Sqh. S2 cells were washed thrice with cold PBS, fixed with 4% EM grade formaldehyde in PBS for 15 min and blocked in 5% BSA in PBS-T (0.1% Triton-X100) for 45min. Cells were then incubated with primary antibodies at RT for 2 hr before proceeding with Duolink PLA (Sigma-Aldrich) according to manufacturer's protocol. After primary antibodies incubation, S2 cells were incubated with PLA probes at 37°C for 1 hr. Cells were washed twice with Buffer A for 5 min each at RT, followed by ligation of probes at 37°C for 30 min. Amplification was performed at 37°C for 100 min, followed by two washes with Buffer B for 10 min each at RT. Cells were washed once with 0.01x Buffer B before incubating with primary antibodies diluted in 3% BSA in PBS for 2 hr at RT.

Cells were washed twice with 0.1% PBS-T and incubated with secondary antibodies for 1.5 hr at RT before mounted with the provided in situ mounting media with DAPI (Duolink, Sigma-Aldrich).

## Acknowledgements

We thank P Somerharju AC Martin, JA Brill, D Glover, C Doe, M Gatti, RE Ward IV, R Karess, S Bogdan, L Alphey, A Chiba, D W Tank, R Ward, F Matsuzaki, A Chiba, T Lee, J Knoblich, J Skeath, A Wodarz, E Schejter, the Bloomington Drosophila Stock Center, Vienna Drosophila Resource Center, Drosophila Genomics Resource Center and the Developmental Studies Hybridoma Bank for fly stocks, pyrene-labeled phospholipids, ESTs and antibodies. We thank F Yu for providing the collection of EMS mutants from which $vib^{133}$ and $vib^{1105}$ were isolated. This work is supported by Ministry of Education Tier 2 MOE2016-T2-2-042 (HW, CTK). ML, SKH and VAB are supported by grants RO1GM112591 and BE-0017 from the National Institutes of Health and the Robert A. Welch Foundation to VAB, respectively.

## Additional information

### Funding

| Funder | Grant reference number | Author |
|---|---|---|
| Ministry of Education - Singapore | MOE2016-T2-2-042 | Chwee Tat Koe Hongyan Wang |
| National Institutes of Health | RO1GM112591 | Max Lönnfors Seong Kwon Hur Vytas A Bankaitis |
| Welch Foundation | BE-0017 | Max Lönnfors Seong Kwon Hur Vytas A Bankaitis |

The funders had no role in study design, data collection and interpretation, or the decision to submit the work for publication.

### Author contributions

Chwee Tat Koe, Conceptualization, Supervision, Funding acquisition, Writing—original draft, Project administration, Writing—review and editing; Ye Sing Tan, Conceptualization, Data curation, Formal analysis, Validation, Investigation, Methodology, Writing—original draft, Writing—review and editing; Max Lönnfors, Data curation, Formal analysis, Validation, Methodology; Seong Kwon Hur, Christine Siok Lan Low, Data curation, Investigation, Methodology; Yingjie Zhang, Data curation, Methodology; Pakorn Kanchanawong, Data curation, Methodology, Generated a plasmid that is used for this study; Vytas A Bankaitis, Supervision, Methodology; Hongyan Wang, Conceptualization, Supervision, Methodology, Writing—review and editing

### Author ORCIDs

Chwee Tat Koe http://orcid.org/0000-0003-0940-125X
Hongyan Wang http://orcid.org/0000-0003-4623-1878

### Decision letter and Author response

Decision letter https://doi.org/10.7554/eLife.33555.025
Author response https://doi.org/10.7554/eLife.33555.026

## Additional files

### Supplementary files

• Transparent reporting form
DOI: https://doi.org/10.7554/eLife.33555.022

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
