## [Decision Letter]

[Editors’ note: a previous version of this study was rejected after peer review, but the authors submitted for reconsideration. The first decision letter after peer review is shown below.]

Thank you for submitting your work entitled "Vibrator and PI4KIIIα Govern Neuroblast Polarity by Anchoring Non-muscle Myosin II" for consideration by *eLife*. Your article has been reviewed by three peer reviewers, one of whom, Yukiko M Yamashita is a member of our Board of Reviewing Editors and the evaluation has been overseen by a Senior Editor. The following individuals involved in review of your submission have agreed to reveal their identity: Clemens Cabernard (Reviewer #2).

Our decision has been reached after consultation between the reviewers. Based on these discussions and the individual reviews below, we regret to inform you that your work will not be considered further for publication in *eLife*.

All reviewers agreed that the paper is potentially interesting, as it investigates an unexplored area of asymmetric stem cell division, and thus the enthusiasm on the topic was high. However, at this point, the reviewers felt that specific data do not cleanly support the authors' claim (as detailed in the individual comments below). In particular, the existing gap between Myosin localization to spindle orientation/asymmetric neuroblast division was considered to be a weakness of the current manuscript. Therefore, per *eLife*'s policy to invite revision only when it is expected to be acceptable after a minor revision, we cannot offer publishing your paper at *eLife*.

Reviewer #1:

Wang and colleagues identified vibrator (PITP) as a novel regulator of neuroblast polarity and division. They show that Vib is required for apical localization of Par6 and aPKC, basal localization of Mira, Pros, and Numb (without affecting the localization of Baz, Insc, Pins). Vib is also required for spindle orientation of the neuroblast. They show that Vib regulates the localization of non-muscle myosin II, Sqh, which is known to regulate neuroblast polarity in embryonic neuroblasts.

This is a well-done study to show the role of PI4P and its regulators in the asymmetric division of the *Drosophila* neuroblast. The relationship between PIPs and cortical domain/polarity itself is not novel, but its involvement in the *Drosophila* neuroblast has not been shown (to my knowledge). Also, the authors provide thorough characterization of how PI4P regulation (via function of Vib) contributes to known mechanism(s) of neuroblast polarity. Although I'd like to see the comments from other reviewers (better experts in the field), I think this is a good paper and worth publishing in *eLife*.

Reviewer #2:

Asymmetric cell division generates cell diversity. Stem cells utilize this mechanism to self-renew itself while forming a differentiating sibling at the same time. The role and function of phosphatidolinositol lipids during asymmetric stem cell division is underexplored. Koe et al., characterize a new phosphatidylinositol transfer protein (PITP) called Vibrator/Giotto in asymmetrically dividing *Drosophila melanogaster* neural stem cells (neuroblasts). Vibrator/Giotto has been previously shown to be involved in cytokinesis. The authors propose that Vibrator is regulating asymmetric neuroblast divisions by anchoring non-muscle Myosin II's regulatory light chain (Spaghetti Squash in flies) to the neuroblast cortex. Based on a bimolecular fluorescence complementation assay, Koe et al., propose that Vibrator and Sqh interact in vivo and in vitro. Furthermore, the authors show that depletion of PI4KIIIalpha, a PI4 kinase, essential for PI(4)P synthesis, is also required for cortical Sqh localization and asymmetric cell division. Based on this data, the authors propose that Vibrator and PI4KIIIalpha promote the synthesis of a PI(4)P plasma membrane pool, which might be required for anchoring Myosin to the cell cortex.

Overall, the role and function of phosphatidolinositol lipids during asymmetric cell division and stem cell self-renewal would be of general interest. However, the connection between cortical Sqh localization, neuroblast polarity, spindle orientation and the reported neuroblast "over-growth" and homeostasis phenotypes remains unexplained. Both Vib and PI4KIIIalpha seem to compromise the localization of cortical Myosin but how cortical Myosin affects neuroblast polarity is underexplored in the manuscript by Koe et al. It seems to me that many of the reported neuroblast phenotypes cannot be understood without further exploring this connection. Also, since mutants, failing to localize Myosin to the cortex often do not enter anaphase, or are compromised in completing cytokinesis, I have difficulties to reconcile this observation with the here reported neuroblast phenotypes.

1) The authors report that vib133 type I and type II neuroblast MARCM clones show a neuroblast over-growth phenotype. What exactly is meant by this? Control clones usually contain 1 neuroblast/clone but vib133 clones seem to contain at least two neuroblasts in the pictures shown in Figure 1. It seems to me that the term "neuroblast over-growth" does not apply here since it is more of a lack of differentiation phenotype.

2) Based on clonal analysis, the authors conclude that Vib is required for neuroblast homeostasis in type I and II lineages. Since Vib has been shown to be involved in cytokinesis – a phenotype the authors have also observed – I wonder whether they don't over interpret their clonal analysis data. For instance, it is possible that cytokinesis failure could form two Dpn positive nuclei in the same cytoplasm. Their pictures do not necessarily rule out this possibility. This point must be clarified.

3) Vib133 mutant clones either show an increase in neuroblast number, show an apparent neuroblast overgrowth (see above), lose neuroblasts and also have cytokinesis defects. How do the authors reconcile all these clonal phenotypes with the polarity and spindle orientation defects?

4) The authors claim that Vib::Venus is slightly enriched at the cleavage furrow in telophase neuroblasts. I suggest the authors support this claim with pixel intensity measurements since the provided images don't convince me that there is an increase at the furrow; Vib::Venus signal seems to be stronger also on the forming GMC cortex compared to the apical neuroblast cortex, which could mainly be due to cell-cell contacts.

5) How do the authors explain the observed spindle orientation phenotype? Is Mud mislocalized in *vib* mutant neuroblasts? Did the authors consider spindle architecture (e.g. loss or fewer astral MTs) or centrosomal defects?

6) The reference (Cabernard et al., 2010) should not be used in the context of Sqh::GFP restriction at the cleavage furrow in neuroblasts; Barros et al., reported the localization of Sqh::GFP before the cited study already.

7) The authors conclude that Vib is required for the localization of phosphorylated Sqh to the cell cortex. This conclusion is derived from the observation that Sqh1P staining in *vib* mutant neuroblasts is predominantly cytoplasmic. However, whether this cytoplasmic signal indeed corresponds to phosphorylated but mislocalized Sqh is unclear. Alternatively, it could simply be background signal. Thus, the alternative hypothesis that Vib is necessary for the correct localization and activation of *Drosophila* Rho kinase (Rock) should be tested.

8) The authors provide evidence that lipid binding and transfer activity is critical for neuroblast polarity and asymmetric cell division. However, given the previous Myosin localization data, the authors should test how this relates to Myosin localization. In other words, how is Sqh::GFP and Sqh1P localized when lipid binding and/or transfer activity are abolished.

9) The apparent loss of PI(4)P-GFP at the cleavage furrow should be verified with pixel intensity measurements.

10) The authors suggest that Vib directly interacts with Sqh in vivo based on a BiFC assay. This finding might be true but I am not convinced about the specificity of this signal. Potential pitfalls and drawbacks of this method should thus be stated and an alternative method to verify the interaction should be considered.

11) Similar to *vib* mutant neuroblasts, it should also be tested whether PI4KIII is involved in the localization and/or activation of Rock.

12) I do not see the benefit of pharmacological inhibition of PI4KIII. It does not seem to bring new mechanistic insight and appears rather redundant with mutant analysis. Thus, the authors might consider removing or at least shortening this section.

Reviewer #3:

The study by Koe et al., describes a role for the type I phosphatidylinositol transfer protein Vibrator (Vib) and PI(4)P in promoting cortical localization of non-muscle myosin II regulatory light chain Spaghetti-squash (Sgh). In the absence of either vib or PI4KIIIα function, Sqh fails to properly localize to the cortex leading to defects in the establishment of cortical cell polarity in mitotic neuroblasts. While telophase rescue restores asymmetric localization of apical proteins, basal proteins fail to re-localize to the budding daughter, resulting in ectopic neuroblast formation. The discovery made in this study is potentially interesting but seems to just begin to unravel the insight into how specific lipid modifications can exert a regulatory role in controlling cortical cell polarity. Furthermore, some of the phenotypes shown in the mutant analyses appear to be extremely muddy, and it remains difficult for me to reconcile the reported polarity defects and the outcome of asymmetric neuroblast division. The wide spectrum of polarity defects exhibited by the vib or PI4KIIIα mutant neuroblasts potentially suggest that there are multiple additional downstream defects in addition to mis-regulation of apical-basal protein localization. As such, I am reluctant to support the publication of this study at its current state.

Major concerns:

1) The images showing the *vib* mutant neuroblasts throughout the figures appear extremely inconsistent. For example, the *vib^133^*type II neuroblasts shown in Figure 1 appears to significantly smaller than *vib^133^*type II neuroblasts shown in Figure 4—figure supplement 1. Why are these mutant neuroblasts so much smaller than wild-type neuroblasts? Does Vib play a role in regulating cell size? In addition, the morphology of the *vib^133^*type II neuroblasts shown in the second column of Figure 4—figure supplement 1 appear to suggest that they might be polyploidy. Does Vib play a role in regulating cell division?

2) It is perplexing to me why loss of vib function simultaneously leads to ectopic neuroblast phenotype and neuroblast loss phenotype? My understanding of this system is that loss of basal proteins such as Numb and Brat always leads to ectopic neuroblasts.

3) The authors used "% of marker localization" Figure 4 and Figure 5. It is unclear to me how this quantification is obtained. Please specify.

4) The BiFC image shown in Figure 6 is fairly underwhelming and not very convincing. In addition, the image shown in Figure 6—figure supplement 1 appears to suggest the interaction might occur in the cytoplasm?

[Editors’ note: what now follows is the decision letter after the authors submitted for further consideration.]

Thank you for submitting your article "Vibrator and PI4KIIIα Govern Neuroblast Polarity by Anchoring Non-muscle Myosin II" for consideration by *eLife*. Your article has been reviewed by three peer reviewers, one of whom, Yukiko M Yamashita is a member of our Board of Reviewing Editors and the evaluation has been overseen by K VijayRaghavan as the Senior Editor. The following individuals involved in review of your submission have agreed to reveal their identity: Clemens Cabernard (Reviewer #3).

The reviewers have discussed the reviews with one another and the Reviewing Editor has drafted this decision to help you prepare a revised submission.

This is a revised manuscript of previously rejected manuscript by Wang and colleagues. The authors investigate the relationship between phosphatidylinositol lipids and asymmetric cell division. The authors show that loss of a phosphatidylinositol transfer protein (PITP), called Vibrator, affects neuroblast polarity and Myosin localization. The authors propose that Vibrator is regulating asymmetric neuroblast divisions by anchoring non-muscle Myosin II's regulatory light chain (Spaghetti Squash in flies) to the neuroblast cortex through a direct interaction between Sqh and Vibrator. Finally, the authors propose that the PI4KIIIalpha, a PI4 kinase, essential for PI(4)P synthesis, is involved in anchoring Myosin to the neuroblast cortex, thereby regulating neuroblast polarity and asymmetric cell division.

Reviewers agreed that the manuscript has improved considerably over the last version and most of straightforward issues have been taken care of. With that said, they found that the major issues remained unresolved, making them 'cautiously supportive' of the publication. Consensus concern is that the phenotypes of *vib* mutant might not be explained by the model proposed by the authors, and uncertainties around their model must be resolved experimentally or by more cautious writing.

Major issues:

The pleiotrophy of the vib phenotype and its involvement in asymmetric cell division and neuroblast self-renewal remain unexplained. The newly added data only partially explain why *vib* mutant clones can both have ectopic neuroblasts but also lose neuroblasts, especially in light of the observed cellular phenotypes (see also below). Also, *vib* mutant's cytokinesis defect obscures the interpretation. We suggest to use the over-expression of Vib[T63E) in *vib^133/j5A6^*neuroblasts as the starting point. If Vib[T63E] indeed rescues the cytokinesis defect as they claimed, they should be able to more cleanly figure out the true polarity defects that lead to either ectopic neuroblasts or premature loss of neuroblasts in both type I and II lineages.

It is not entirely convincing that premature nuclear localization of Prospero is the cause of neuroblast loss in *vib* mutant clones. First, the level of nuclear Prospero shown in Figure 1—figure complement 2B is so much lower than the Prospero+ cells in the same clone. Second, the authors acknowledged that type II neuroblasts do not express Prospero. These data strongly suggest that yet-to-be-identified factors contribute to premature differentiation of *vib* neuroblasts. If this is the case, the authors should openly acknowledge this possibility, and tone down their interpretation. Third, it seems that polarity is largely restored in telophase – which should actually be more carefully quantified and displayed in a main figure panel – and thus it is difficult to reconcile how neuroblast homeostasis is affected.

Specific comments:

The authors have tried to replace the word "neuroblast overgrowth" with "ectopic neuroblasts" but missed several. Please make the following corrections for consistency.

Figure 4—figure supplement 1 and the figure legend associated with it.

Introduction – The following statements should be modified: "At telophase, nonmuscle Myosin II localizes to the cleavage furrow and regulates basal protein localization (Barros et al., 2003). "The non-muscle myosin II regulatory light chain protein Spaghetti-squash (Sqh) regulates both asymmetric localization of basal cell fate determinants in neuroblasts and cytokinesis of dividing cells (Karess et al., 1991; Barros et al., 2003)."

It is true that in sqh germ line clones, basal cell fate determinants are mislocalized. Similarly, the authors show here that both aPKC and Mira fail to localized to the apical and basal neuroblast cortex, respectively in sqh RNAi expressing neuroblasts. However, the authors should also state that Miranda's basal confinement is due aPKC mediated phosphorylation as shown by Atwood et al. Throughout the manuscript the authors should thus make a distinction between "anchoring" of polarity proteins to the cell cortex and "asymmetric localization" of the investigated polarity proteins. It seems to me, that the polarity proteins cannot be localized asymmetrically without the proper anchoring. Lack of Sqh and Vib seems to predominantly affect the anchoring but will not reveal new insight into the mechanism for polarized localization and subsequent asymmetric segregation of these proteins. Thus, the phenotype is more general and Vib and phosphatidolinositol lipids might not necessarily regulate asymmetric cell division specifically.

Considering recent publications (Roubinet, 2017; Montembault, 2017), the following statement should also be updated or changed: "However, how myosin localization is regulated during asymmetric division remains unknown".

The authors claim that Vib is localized to the cell cortex with a basal enrichment. I am not convinced that the basal enrichment is specific for Vib; Figure 1—figure supplement 1 and E shows that Vib is also enriched on the apical side. It seems that Vib enrichment mostly occurs whenever other cells are in contact with the imaged neuroblast. In our experience, membrane markers tend to show basal enrichment due to GMCs clustered on the basal neuroblast side. Thus, if the authors want to show that basal enrichment is specific, they should generate primary neuroblast cultures expressing Vib::Venus of neuroblasts devoid of any attached GMCs.

The authors conclude that Vib is critical for asymmetric protein localization and segregation in dividing neuroblasts. I think this statement needs to be modified. I agree that the localization of polarity proteins is affected in metaphase. However, all provided images of the basal cell fate determinant Miranda show normal localization and segregation in telophase (despite some Mira signal on the spindle). Thus, I am confused why the authors conclude that Vib is critical for asymmetric protein segregation if Miranda and aPKC are normally localized in many *vib* mutant telophase neuroblasts.

The following statement "the localization of Sqh::GFP at cleavage furrow was lost" and the subsequent conclusions should be modified. Figure 3 shows that in *vib* mutant telophase neuroblasts, Sqh::GFP is still localized at the neuroblast cortex (especially with the 1105 allele) although a pronounced furrow enrichment is lacking. It seems as if the overall distribution of Sqh is changed compared to wild type. However, the provided image still shows it to be localized at the cell cortex.

I still don't understand the nature of vib133's pleiotropic phenotype. Vib133 mutant clones either show an increase in neuroblast number, lose neuroblasts and have cytokinesis defects.

I appreciate the new data showing that expression of Pros could lead to differentiation of *vib* mutant neuroblasts. However, how can that phenotype be explained considering the observed polarity defects? Loss of aPKC usually causes loss of neuroblasts and not an increase in Nb numbers. Similarly, I am unconvinced that the observed spindle orientation phenotype could result in ectopic neuroblasts; to this end, the authors should show telophase figures, demonstrating that misaligned metaphase spindles indeed give rise to symmetric neuroblast divisions, producing siblings with neuroblast fate. Given the relatively weak nature of the spindle orientation defects, I am not convinced that the authors could find this.

I am not an expert on the lipid strip assay, however, I feel that a positive control is missing in this experiment (Figure 5).

I am further confused by the role of PI(4)P in cortical Sqh localization. If PI(4)P is essential for cortical Sqh localization – which the authors claim is mostly lost in *vib* mutants – why is the localization of PI(4) in *vib* mutants only affected at the cleavage furrow in *vib* mutants? Overall, neuroblasts still contain robust PI(4)P localization in *vib* mutants. The same is also true for sqh mutants, which still contain robust levels of PI(4)P at the neuroblast membrane.

[Editors' note: further revisions were requested prior to acceptance, as described below.]

Thank you for resubmitting your article "Vibrator and PI4KIIIα Govern Neuroblast Polarity by Anchoring Non-muscle Myosin II" for consideration by *eLife*. Your article has been reviewed by the same reviewers as former submissions, and the evaluation has been overseen by K VijayRaghavan as the Senior Editor.

All reviewers have agreed that your work provides important new insights into asymmetric division of the *Drosophila* neuroblasts and thus merits publication in *eLife*.

However, before we can formally accept your paper, we would like you to make following textual edits. After this modification, we will be able to accept your manuscript editorially.

1) The reviewers felt that your explanation of 'the ratio of segregated apical vs. basal fate determinant' was not satisfying, especially because no data are offered to back up this notion. Thus, as of now, this explanation is hand-waving. Although the authors do not need to add more experiments and quantification (unless it is easily possible by analyzing existing data), the authors should acknowledge this shortcoming in interpreting their data.

2) Also, now that the authors do *not* claim that Vib is concentrated on basal side and state that Vib is (early) evenly cortical, it clearly leaves the readers wondering what Vib might be doing at the apical side. The reviewer acknowledges that testing 'localization-specific function' is difficult for anybody to do, and do not ask the authors to conduct such experiments: however, again, the authors should more carefully discuss about this, and at least propose possible models.

3) Also, the authors claim that "We have already included the quantification of telophase localization of both aPKC and Mira in *vib^1105/j5A6^* and *vib^133/j5A6^*telophase neuroblasts in our previous manuscript (Figure 2 and Figure 2—figure supplement 1)".

I can find the telophase data for Mira and Pros in Figure 4 and Figure 4—figure supplement 2 but not for aPKC. Maybe it was present in an earlier version and was since removed. If so, it would make sense to include it again.

---

## [Author Response]

[Editors’ note: what now follows is the decision letter after the authors submitted for further consideration.]

Reviewer #1:

Wang and colleagues identified vibrator (PITP) as a novel regulator of neuroblast polarity and division. They show that Vib is required for apical localization of Par6 and aPKC, basal localization of Mira, Pros, and Numb (without affecting the localization of Baz, Insc, Pins). Vib is also required for spindle orientation of the neuroblast. They show that Vib regulates the localization of non-muscle myosin II, Sqh, which is known to regulate neuroblast polarity in embryonic neuroblasts.This is a well-done study to show the role of PI4P and its regulators in the asymmetric division of the Drosophila neuroblast. The relationship between PIPs and cortical domain/polarity itself is not novel, but its involvement in the Drosophila neuroblast has not been shown (to my knowledge). Also, the authors provide thorough characterization of how PI4P regulation (via function of Vib) contributes to known mechanism(s) of neuroblast polarity. Although I'd like to see the comments from other reviewers (better experts in the field), I think this is a good paper and worth publishing in eLife.

We thank the review for her positive endorsement on our manuscript.

Reviewer #2:Overall, the role and function of phosphatidolinositol lipids during asymmetric cell division and stem cell self-renewal would be of general interest. However, the connection between cortical Sqh localization, neuroblast polarity, spindle orientation and the reported neuroblast "over-growth" and homeostasis phenotypes remains unexplained. Both Vib and PI4KIIIalpha seem to compromise the localization of cortical Myosin but how cortical Myosin affects neuroblast polarity is underexplored in the manuscript by Koe et al. It seems to me that many of the reported neuroblast phenotypes cannot be understood without further exploring this connection. Also, since mutants, failing to localize Myosin to the cortex often do not enter anaphase, or are compromised in completing cytokinesis, I have difficulties to reconcile this observation with the here reported neuroblast phenotypes.

We thank the review for finding the topic of our study interesting and his constructive comments. In the revised manuscript, we provide new data showing that Sqh regulates PI(4)P localization on the plasma membrane in neuroblasts (subsection “Sqh binds to phosphoinositide PI(4)P and localizes it to cell cortex in neuroblasts”). We analysed the localization of PI(4)P-GFP in *sqh1* hemizygotes, a hypomorphic allele of Sqh that survived to the third instar larval stage. Remarkably, 23.3% of interphase neuroblasts (n=313) and 36.8% of metaphase neuroblasts (n=19) displayed distinct PI(4)P-GFP aggregates in the cytoplasm (Figure 5). Knocking down *sqh* using *insc*-Gal4 also resulted in the formation of many distinct PI(4)P-GFP aggregates in interphase neuroblasts (Figure 5, 30%, n=40). Thus, Sqh likely regulates neuroblast polarity by facilitating the membrane localization of PI(4)P.

1) The authors report that vib133 type I and type II neuroblast MARCM clones show a neuroblast over-growth phenotype. What exactly is meant by this? Control clones usually contain 1 neuroblast/clone but vib133 clones seem to contain at least two neuroblasts in the pictures shown in Figure 1. It seems to me that the term "neuroblast over-growth" does not apply here since it is more of a lack of differentiation phenotype.

The reviewer’s point is well taken. We have replaced the term of “neuroblast overgrowth” with “ectopic neuroblasts” throughout the manuscript to precisely describe the phenotype of *vib* mutant clones.

2) Based on clonal analysis, the authors conclude that Vib is required for neuroblast homeostasis in type I and II lineages. Since Vib has been shown to be involved in cytokinesis – a phenotype the authors have also observed – I wonder whether they don't over interpret their clonal analysis data. For instance, it is possible that cytokinesis failure could form two Dpn positive nuclei in the same cytoplasm. Their pictures do not necessarily rule out this possibility. This point must be clarified.

In our analysis on neuroblast homeostasis in clones, neuroblasts with obvious cytokinesis defects (polyploidy cells) were excluded from our analysis. Neuroblasts with two (Dpn+) nuclei in the same cytoplasm were also excluded in our analysis. We only counted ectopic neuroblasts that were clearly separated cells based on CD8-GFP, which marked the outline of cells. We have updated our quantification methods in the revised manuscript (Discussion section). In addition, we have replaced the images that better represent our quantification (Figure 1, *vib^1105^*clone and Figure 1—figure supplement 1, *vib133*).

In addition, we would like to point out that the expression of VibT63E::Venus rescued the cytokinesis defect in 98.7% of *vib133* clones (n=75), while excess neuroblasts persisted in these clones (Figure 4, Figure 4—figure supplement 1; subsection “The lipid binding and transfer activity of Vib is critical for asymmetric division and homeostasis of neuroblasts”). This data suggests that ectopic neuroblasts observed in *vib* mutants are not due to cytokinesis failure.

3) Vib133 mutant clones either show an increase in neuroblast number, show an apparent neuroblast overgrowth (see above), lose neuroblasts and also have cytokinesis defects. How do the authors reconcile all these clonal phenotypes with the polarity and spindle orientation defects?

Vib has a known function in cytokinesis, so it is not surprising to observe the cytokinesis defect in *vib-* neuroblasts. Besides cytokinesis defect, we report in this manuscript that *vib133* mutant also displayed phenotype of ectopic neuroblasts and loss of neuroblasts. We propose that the ectopic neuroblasts were due to a disruption of neuroblast polarity and spindle orientation defects. To explore the mechanism underlying loss of neuroblasts in *vib*mutants, we first examined cell death with active Caspase-3 staining and found that *vib*neuroblasts did not undergo cell death, similar to wild-type control (Figure 1—figure supplement 2). Next, we examined whether loss of neuroblasts was due to premature differentiation of neuroblasts. To this end, we examined the localization of Pros, a differentiation factor, in neuroblast lineages. Whereas none of type I neuroblast control clones had nuclear Pros expression (n=15), 66.7% of type I *vib133* neuroblast clones expressed nuclear Pros (n=12) (Figure 1—figure supplement 2). Furthermore, we examined *vib^133/j5A6^*and *vib^133/j5A6^*trans-heterozygous neuroblasts. While none of control neuroblasts showed nuclear Pros expression (n=121), in *vib^133/j5A6^*and *vib^133/j5A6^*trans-heterozygotes brains, 50% (n=158) and 56.8% (n=243) of interphase neuroblasts exhibited nuclear Pros expression (Figure 1—figure supplement 2). Therefore, loss of *vib* could disrupt Pros localization, resulting in earlier termination of proliferation as well as premature differentiation and hence loss of neuroblasts in some of the *vib133* mutants. We have included these results

in Figure 1—figure supplement 2, subsection “Vib controls homeostasis of larval central brain neuroblasts”.

4) The authors claim that Vib::Venus is slightly enriched at the cleavage furrow in telophase neuroblasts. I suggest the authors support this claim with pixel intensity measurements since the provided images don't convince me that there is an increase at the furrow; Vib::Venus signal seems to be stronger also on the forming GMC cortex compared to the apical neuroblast cortex, which could mainly be due to cell-cell contacts.

To determine if Vib-Venus is enriched at the cleavage furrow, we measured the pixel intensity of Venus on the cell cortex of wild-type telophase neuroblasts and calculated the ratio of Vib-Venus intensity at the cleavage furrow to that at the apical cortex (termed VenusCF: VenusApical). Wild-type telophase neuroblasts had the average ratio of 1.33 (n=33) with 9% of telophase neuroblasts showed 2 times enrichment at the cleavage furrow. These data suggest that there is slight enrichment of Vib-Venus at the basal side and cleavage furrow of neuroblasts. We have included these results in Figure 1—figure supplement 1, subsection “Vib controls homeostasis of larval central brain neuroblasts”.

5) How do the authors explain the observed spindle orientation phenotype? Is Mud mislocalized in vib mutant neuroblasts? Did the authors consider spindle architecture (e.g loss or fewer astral MTs) or centrosomal defects?

We showed in our initial manuscript that Gαi was partially delocalized in *vib-* neuroblasts Figure 2—figure supplement 1). This likely contributed to the spindle orientation phenotype in *vib-* mutants. We examined Mud localization in *vib-* metaphase neuroblasts and found that centrosomal Mud localization was largely unaffected (Figure 2—figure supplement 1; 96.3%, n=27; Control, n=23). Unfortunately, the anti-Mud antibody we previously obtained from Dr. Fumio Matsuzaki was finished and the new aliquot obtained from his lab only decorated centrosomes but not apical cortex of neuroblasts. Thus, we were unable to directly determine whether the apical/cortical localization of Mud was delocalized in *vib*mutants. Nevertheless, given that Pins directly interacts with Mud and localizes Mud to apical/cortical neuroblasts (Bowman et al., 2006; Izumi et al., 2006; Siller et al., 2006) and that Pins is predominantly apically localized in *vib* loss-of-function mutants (Figure 2—figure supplement 1), apical/cortical localization of Mud is unlikely to be disrupted in *vib*neuroblasts. Centrosomes in *vib-* neuroblasts were unaffected, as Cnn localization is normal in both control (n=24) and *vib^133/j5A6^*neuroblasts (n=47) (Figure 2—figure supplement 1). Likewise, the spindle architecture is unaffected in *vib^133/j5A6^*neuroblasts (Figure 2—figure supplement 1; n=43; Control, n=25). They were able to assemble normal-looking mitotic spindle and astral microtubules that were labeled by α-tubulin. Taken together, GαI delocalization, but not spindle architecture or centrosomal defects, most likely caused the spindle orientation defects observed in *vib* mutants. We have included these results in Figure 2—figure supplement 1, subsection “Vib is important for apical-basal polarity and spindle orientation in neuroblasts”.

6) The reference (Cabernard et al., 2010) should not be used in the context of Sqh::GFP restriction at the cleavage furrow in neuroblasts; Barros et al., reported the localization of Sqh::GFP before the cited study already.

Done.

7) The authors conclude that Vib is required for the localization of phosphorylated Sqh to the cell cortex. This conclusion is derived from the observation that Sqh1P staining in vib mutant neuroblasts is predominantly cytoplasmic. However, whether this cytoplasmic signal indeed corresponds to phosphorylated but mislocalized Sqh is unclear. Alternatively, it could simply be background signal. Thus, the alternative hypothesis that Vib is necessary for the correct localization and activation of Drosophila Rho kinase (Rock) should be tested.

(Rok) that phosphorylates and activates Sqh, we first determined the localization of Rok-GFP in *vib* mutant neuroblasts. In controls (Ubi-Rok-GFP heterozygous), neuroblasts expressed cortical Rok-GFP at metaphase (93.5%, n=46) and showed enriched Rok-GFP at the cleavage furrow at telophase (91.7%, n=24). Similarly, in *vib^133/j5A6^*we observed cortical Rok-GFP localization in 91.6% of metaphase neuroblasts (n=24) and 85.7% of telophase neuroblasts showed enriched Rok-GFP at the cleavage furrow (n=14) (Figure 3—figure supplement 1, subsection “Vib anchors non-muscle myosin II regulatory light chain Spaghetti-squash (Sqh) to the cell cortex in neuroblasts”). This result suggests that Vib is not essential for Rok localization in neuroblasts.

Next, we explored whether increase Rok activity could rescue the asymmetric cell division defects observed in *vib* mutant. We overexpressed the catalytic domain of Rok (Rok-CAT), a previously characterized transgene (Winter et al., 2001) could rescue Mira delocalization in *vib^133/j5A6^*. Expression of RokCAT in wild type neuroblasts had no influence on Mira localization in neuroblasts (Figure 3—figure supplement 1=27 for metaphase; n=13 for telophase). We found that ectopic expression of the RokCAT in *vib^133/j5A6^*did not significantly rescue Mira delocalization in neuroblasts (metaphase, 85.7%, n=42; telophase, 83.3%, n=12), rather showing similar Mira delocalization to *vib^133/j5A6^*neuroblasts (metaphase, 90.2%, n=34; telophase, 84.6%, n=26). These observations suggested that Vib regulates Sqh localization unlikely via localizing/activating Rok. We have included these new data in Figure 3—figure supplement 1 and subsection “Vib anchors non-muscle myosin II regulatory light chain Spaghetti-squash (Sqh) to the cell cortex in neuroblasts”.

8) The authors provide evidence that lipid binding and transfer activity is critical for neuroblast polarity and asymmetric cell division. However, given the previous Myosin localization data, the authors should test how this relates to Myosin localization. In other words, how is Sqh::GFP and Sqh1P localized when lipid binding and/or transfer activity are abolished.

To test if the lipid transfer activity of Vib is crucial for the localization of Sqh, we first examined the localization of Sqh in *vib^133/j5A6^*transheterozygotes expressing either VibT63A or VibT63E. In wild-type controls, 81.3% (n=32) of metaphase neuroblasts exhibited cortical Sqh localization and 84.6% (n=13) of telophase neuroblasts enriched Sqh at the cleavage furrow (Figure 4—figure supplement 2). Expression of either VibT63A::Venus (metaphase, n=37; telophase, n=19) or VibT63E::Venus (metaphase, n=36; telophase, n=20) in wild-type controls had no significant effect on Sqh localization in neuroblasts (Figure 4—figure supplement 2). Consistent with earlier observations (Figure 3—figure supplement 1), 82.9% (n=41) of *vib^133/j5A6^*metaphase neuroblasts showed cytoplasmic Sqh localization and 64.3% of telophase neuroblasts (n=14) failed to enrich Sqh at the cleavage furrow (Figure 4—figure supplement 2). Similarly, 88.1% (n=59) of metaphase neuroblasts and 71.4% (n=21) of telophase neuroblasts in *vib^133/j5A6^*expressing VibT63A::Venus showed the delocalization of Sqh (Figure 4—figure supplement 2). In addition, 88.4% (n=43) of *vib^133/j5A6^*metaphase neuroblasts expressing VibT63E::Venus displayed cytoplasmic Sqh localization and 53.3% (n=15) of telophase neuroblasts failed to enrich Sqh at the cleavage furrow (Figure 4—figure supplement 2).

Next, we examined the localization of Sqh1P in various genotypes. In wild-type controls, 88.5% (n=87) of metaphase neuroblasts exhibited cortical Sqh1P localization and 91.2% (n=34) of telophase neuroblasts showed localization of Sqh1P at the cleavage furrow (Figure 4—figure supplement 2). Sqh1P localization in wild-type neuroblasts expressing VibT63A::Venus (metaphase, n=54; telophase, n=22) or VibT63E::Venus (metaphase, n=45; telophase, n=21) was similar to that of wild-type controls (Figure 4—figure supplement 2;

quantifications are in figure legends). By contrast, 82.3% of *vib^133/j5A6^*metaphase neuroblasts (n=62) showed cytoplasmic Sqh1P localization and 50% of telophase neuroblasts (n=20) failed to enrich Sqh1P at the cleavage furrow (Figure 4—figure supplement 2). Similar to Sqh1P delocalization in *vib^133/j5A6^*mitotic neuroblasts, 80% (n=65) of *vib^133/j5A6^*metaphase neuroblasts with VibT63A::Venus were cytoplasmic and only 45% (n=20) of telophase neuroblasts were localized at the cleavage furrow. Likewise, 85.7% (n=70) of *vib^133/j5A6^*neuroblasts expressing VibT63E::Venus displayed cytoplasmic Sqh1P in metaphase and only 38.5% (n=26) of neuroblasts had Sqh1P localization at the cleavage furrow in telophase (Figure 4—figure supplement 2). These observations suggest that Sqh1p is delocalized when lipid binding and transfer activity of Vib are abolished. We have included these data in Figure 4—figure supplement 2, subsection “The lipid binding and transfer activity of Vib is critical for asymmetric division and homeostasis of neuroblasts.”

9) The apparent loss of PI(4)P-GFP at the cleavage furrow should be verified with pixel intensity measurements.

We measured the PI(4)P-GFP intensity at the cleavage furrow of telophase neuroblasts. Normalized against the intensity at apical cortex, wild-type neuroblasts showed an average of 2.07 fold enrichment (n=34) of PI(4)P-GFP at the cleavage furrow, while in *vib^133/j5A6^*telophase neuroblasts, the enrichment of PI(4)P-GFP decreased to 1.44 fold (n=18) at the cleavage furrow (Figure 5). There was also a decrease of basal enrichment of PI(4)P-GFP at basal cortex of *vib^133/j5A6^* neuroblasts, from 52.9% (average of 1.56 fold, n=34) of wild-type neuroblasts to 33.3% (average of 1.26 fold, n=18) of *vib^133/j5A6^* neuroblasts (Figure 5). We have included these new data in Figure 5 and subsection “Sqh binds to phosphoinositide PI(4)P and localizes it to cell cortex in neuroblasts”.

10) The authors suggest that Vib directly interacts with Sqh in vivo based on a BiFC assay. This finding might be true but I am not convinced about the specificity of this signal. Potential pitfalls and drawbacks of this method should thus be stated and an alternative method to verify the interaction should be considered.

To validate the interaction between Vib and Sqh in BiFC, we first generated additional important negative controls, two truncated Vib proteins, Vib N-terminus (VibN) and Vib C- terminus (VibC), and tested either truncated form of Vib abolished the interaction with Sqh in BiFC assays. Indeed, neither VibN-Myc-NYFP nor VibC-Myc-NYFP interacted with CYFPHA-Sqh (Figure 6—figure supplement 1). These results further validated the interaction between Vib and Sqh in BiFC assay (subsection “Vib interacts with Sqh in vitroand in vivoin BiFC assays”).

Next, we improved in vivo BiFC data by generating transgenic flies co-expressing NYFP-Myc-Vib with CYFP-HA to replace NYFP-Myc-Vib single expression control. Likewise, we have also generated transgenic flies co-expressing and replaced the single expression control CYFP-HA-Sqh with the new control co-expressing both NYFP-Myc and CYFP-HASqh. Based on the new controls, we still observed specific YFP signal when co-expressing NYFP-Myc-Vib and CYFP-HA-Sqh (Figure 6, subsection “Vib interacts with Sqh in vitroand in vivoin BiFC assays”).

Furthermore, we took an alternative approach and performed a proximity ligation assay (PLA). PLA allows detection and visualization of protein-protein interactions with high specificity and sensitivity in the form of fluorescence signal in situ (Fredriksson et al., 2002). We co-stained S2 cells with anti-Flag and anti-Myc antibodies to visualize the co-expression of proteins tagged with Flag or Myc. Quantifications of PLA foci were carried out only in cells with both Myc and Flag signals (Figure 6). In S2 cells expressing Flag and Myc controls, we did not observe any PLA fluorescence signal (Figure 6=96). In S2 cells that were co-transfected with Flag-Vib and Myc controls, 76.6% cells (n=188) had no PLA signal and 19.2% cells displayed weak PLA fluorescence signal (≤10 foci), with the remaining 3.7% and 0.5% cells displaying PLA foci between 11-30 and >30, respectively. In control expressing Myc-Sqh with Flag, 86.6% cells (Figure 6=149) had no PLA signal and 13.4% of cells with weak PLA signal (≤ 10 foci) PLA foci. By contrast, 82.9% (Figure 6=216) of S2 cells expressing both Flag-Vib and Myc-Sqh showed PLA fluorescence: 15.3% of cells displayed strong PLA signal (>30 foci), 24.5% of cells showed moderate PLA signal (11-30 foci) and 43.1% of cells showed weak PLA signal (<10 foci). On average, controls Flag-Vib with Myc and Myc-Sqh with Flag resulted in 1.7 and 0.3 PLA foci per cell, respectively. By contrast, co-expressing Flag-Vib and Myc-Sqh resulted in 15.2 PLA foci per cell. These data (subsection “Vib interacts with Sqh in vitroand in vivoin BiFC assays”) reinforced that Vib and Sqh physically interact.

Taken together, we concluded that Vib physically associates with Sqh.

11) Similar to vib mutant neuroblasts, it should also be tested whether PI4KIII is involved in the localization and/or activation of Rock.

Since Rok-GFP cannot be directly examined in PI4KIIIα MARCM clones that are also marked by CD8-GFP, we analysed Rok-GFP localization upon pharmacological inhibition of PI4KIIIα by treating larval brains expressing Ubi-Rok-GFP with 2.5μM PAO for 2 hours. In the mock treatment, 92.7% (n=55) of metaphase neuroblasts exhibited cortical Rok-GFP localization and 70% (n=20) of telophase neuroblasts showed enriched Rok-GFP localization at the cleavage furrow. With PAO treatment, 92% (n=112) of metaphase neuroblasts with cortical Rok-GFP localization and 76.5% (n=17) of telophase with Rok-GFP enriched at the cleavage furrow were observed. This result suggests that Rok localization or activation is likely independent of PI4KIIIα function (Figure 7—figure supplement 2, subsection “PI4KIIIα localizes Sqh to the cell cortex in neuroblasts”).

12) I do not see the benefit of pharmacological inhibition of PI4KIII. It does not seem to bring new mechanistic insight and appears rather redundant with mutant analysis. Thus, the authors might consider removing or at least shortening this section.

We have shortened this section in the revised manuscript (subsection “PI4KIIIα localizes Sqh to the cell cortex in neuroblasts”).

Reviewer #3:The study by Koe et al., describes a role for the type I phosphatidylinositol transfer protein Vibrator (Vib) and PI(4)P in promoting cortical localization of non-muscle myosin II regulatory light chain Spaghetti-squash (Sgh). In the absence of either vib or PI4KIIIα function, Sqh fails to properly localize to the cortex leading to defects in the establishment of cortical cell polarity in mitotic neuroblasts. While telophase rescue restores asymmetric localization of apical proteins, basal proteins fail to re-localize to the budding daughter, resulting in ectopic neuroblast formation. The discovery made in this study is potentially interesting but seems to just begin to unravel the insight into how specific lipid modifications can exert a regulatory role in controlling cortical cell polarity. Furthermore, some of the phenotypes shown in the mutant analyses appear to be extremely muddy, and it remains difficult for me to reconcile the reported polarity defects and the outcome of asymmetric neuroblast division. The wide spectrum of polarity defects exhibited by the vib or PI4KIIIα mutant neuroblasts potentially suggest that there are multiple additional downstream defects in addition to mis-regulation of apical-basal protein localization. As such, I am reluctant to support the publication of this study at its current state.

We thank the reviewer for considering our work interesting and his/her constructive comments. In the revised manuscript, we provide new data showing that Sqh regulates PI(4)P localization on the plasma membrane in neuroblasts (subsection “Sqh binds to phosphoinositide PI(4)P and localizes it to cell cortex in neuroblasts”). We analysed the localization of PI(4)P-GFP in *sqh^1^*hemizygotes, a hypomorphic allele of Sqh that survived to the third instar larval stage. Remarkably, 23.3% of interphase neuroblasts (n=313) and 36.8% of metaphase neuroblasts (n=19) displayed distinct PI(4)P-GFP aggregates in the cytoplasm (Figure 5). In addition, knocking down *sqh* using *insc*-Gal4 resulted in the formation of many distinct PI(4)P-GFP aggregates in interphase neuroblasts (Figure 5, 30%, n=40). Thus, Sqh likely regulates neuroblast polarity by facilitating the membrane localization of PI(4)P.

In the revised manuscript, we have improved the quality of analysis by increasing sample size for many of the experiments.

We showed in this manuscript that *vib* depletion resulted in Sqh delocalization in neuroblast, which likely caused the delocalization of aPKC and Mira. Given that Sqh is required for asymmetric localization of aPKC and Mira, Sqh is an important although not the only downstream regulator of Vib during neuroblast asymmetric division.

Together with new data described below in the point-to-point response, I hope to convince the reviewer that we have significantly improved the manuscript and the revised manuscript is now suitable for publication in *eLife.*

Major concerns:1) The images showing the vib mutant neuroblasts throughout the figures appear extremely inconsistent. For example, the vib^13 3^type II neuroblasts shown in Figure 1 appears to significantly smaller than vib^133^ type II neuroblasts shown in Figure 4—figure supplement 1. Why are these mutant neuroblasts so much smaller than wild-type neuroblasts? Does Vib play a role in regulating cell size? In addition, the morphology of the vib^133^ type II neuroblasts shown in the second column of Figure 4—figure supplement 1 appear to suggest that they might be polyploidy. Does Vib play a role in regulating cell division?

We measured the diameter of *vib^133^* neuroblasts and found that some but not all of mutant neuroblasts showed reduced cell diameter. The average size of *vib^133^*neuroblasts (7.76μm, n=27) is smaller than neuroblasts from control MARCM clones (10.79μm, n=37) (Figure 1—figure supplement 2). We have replaced the image in Figure 1—figure supplement 1 with a more representative image to show the reduced neuroblast size. Next, to investigate whether Vib plays a role in regulating cell size, we measured the ratio of nucleolar to nuclear size in neuroblasts, which is an indicator of cellular growth. With nucleolus: nucleus ratio in wild-type control neuroblasts normalized as 1 (n=29), the ratio in *vib^133^* neuroblasts was reduced slightly to 0.81 (n=28) (Figure 1—figure supplement 2, subsection “Vib controls homeostasis of larval central brain neuroblasts”). This result suggests that cell growth in *vib^-^* mutant neuroblast is mildly reduced. The reduced neuroblast size in some of the neuroblasts may be due to the nuclear localization of Pros (Figure 1—figure supplement 2 subsection “Vib controls homeostasis of larval central brain neuroblasts”), resulting in earlier termination of proliferation and premature differentiation.

Vib has a known role in regulating cell division, in particular cytokinesis (Gatt and Glover, 2006; Giansanti et al., 2006). We also observed cytokinesis defects in neuroblasts and showed the polyploidy cell in Figure 4—figure supplement 1 to faithfully document the phenotypes. Interestingly, VibT63E, with deficient π binding and transfer activity but retain sufficient PC binding and transfer activity, almost completely rescued cytokinesis defects observed in *vib* mutants, but failed to rescue asymmetric division defects (Figure 4, Figure 4—figure supplement 1; subsection “The lipid binding and transfer activity of Vib is critical for asymmetric division and homeostasis of neuroblasts”). Thus, despite the pleotropic phenotypes observed in *vib* neuroblasts, asymmetric division defects is unlikely a consequence of cytokinesis defect and these two phenotypes could be uncoupled in the above experiments.

2) It is perplexing to me why loss of vib function simultaneously leads to ectopic neuroblast phenotype and neuroblast loss phenotype? My understanding of this system is that loss of basal proteins such as Numb and Brat always leads to ectopic neuroblasts.

To explore the mechanism underlying loss of neuroblasts in *vib^-^* mutants, we first examined cell death of neuroblasts by labelling active Caspase-3 and found that neuroblasts in *vib^-^* mutants were not undergoing cell death, similar to wild-type controls (Figure 1—figure supplement 2). Next, we examined whether loss of neuroblasts was due to premature differentiation of neuroblasts. To this end, we examined the nuclear localization of Pros, a differentiation factor, in neuroblasts. Because type II neuroblast do not express Pros, we focused our analysis on type I neuroblasts. Whereas none of type I neuroblast control clones had nuclear Pros (n=15), 66.7% of type I *vib^133^* neuroblast clones expressed nuclear Pros (n=12) (Figure 1—figure supplement 2). Furthermore, we examined *vib^133/j5A6^* and *vib^1105/j5A6^* trans-heterozygous neuroblasts. While none of control neuroblasts showed nuclear Pros expression (n=121), in *vib^133/j5A6^* and *vib^1105/j5A6^* mutants, 50% (n=158) and 56.8% (n=243) of interphase neuroblasts exhibit nuclear Pros, respectively (Figure 1—figure supplement 2). Therefore, loss of *vib* resulted in nuclear Pros localization in neuroblasts, leading to premature differentiation and hence loss of neuroblasts in some of the *vib^133^*mutants. We have included these results in Figure 1—figure supplement 2, subsection “Vib controls homeostasis of larval central brain neuroblasts”.

3) The authors used "% of marker localization" Figure 4 and Figure 5. It is unclear tome how this quantification is obtained. Please specify.

To improve the clarity, we have replaced the Y axis labelling with “% of metaphase Neuroblasts with Mira” in Figure 4 and with “% of Neuroblasts with PI(4)P-GFP” in Figure 5.

4) The BiFC image shown in Figure 6 is fairly underwhelming and not very convincing. In addition, the image shown in Figure 6—figure supplement 1 appears to suggest the interaction might occur in the cytoplasm?

To validate the interaction between Vib and Sqh in BiFC, we first generated additional important negative controls, two truncated Vib proteins, Vib N-terminus (VibN) and Vib C- terminus (VibC), and tested either truncated form of Vib abolished the interaction with Sqh in BiFC assays. Indeed, neither VibN-Myc-NYFP nor VibC-Myc-NYFP interacted with CYFPHA-Sqh (Figure 6—figure supplement 1). These results further validated the interaction between Vib and Sqh in the BiFC assay in S2 cells (subsection “Vib interacts with Sqh in vitroand in vivoin BiFC assays”). Based on our in vivo BiFC data, Vib and Sqh interaction likely occurs on the cell cortex of neuroblasts (Figure 6).

Next, we improved in vivo BiFC assays by generating transgenic flies co-expressing NYFP-Myc-Vib with CYFP-HA to replace NYFP-Myc-Vib single expression control. Likewise, we have also generated transgenic flies co-expressing NYFP-Myc and CYFP-HA-Sqh and replaced the single expression control CYFP-HA-Sqh with the co-expression of NYFP-Myc and CYFP-HA-Sqh. Based on the new controls, we still observed specific YFP signal when co-expressing NYFP-Myc-Vib and CYFP-HA-Sqh, but not the new controls (Figure 6, subsection “Vib interacts with Sqh in vitroand in vivoin BiFC assays”).

Furthermore, we took an alternative approach and performed a proximity ligation assay (PLA). PLA allows detection and visualization of protein-protein interactions with high specificity and sensitivity in the form of fluorescence signal in situ (Fredriksson et al., 2002). We co-stained S2 cells with anti-Flag and anti-Myc antibodies to visualize the co-expression of proteins tagged with Flag or Myc. Quantifications of PLA foci were carried out only in cells with co-expression of Flag and Myc-tagged proteins (Figure 6). In S2 cells expressing Flag and Myc controls, we did not observe any PLA fluorescence signal (Figure 6=96). In S2 cells that were co-transfected with Flag-Vib and Myc controls, 76.6% cells (n=188) had no PLA signal and 19.2% cells displayed weak PLA fluorescence signal (≤10 foci), with the remaining 3.7% and 0.5% cells displaying PLA foci between 11-30 and >30, respectively. In control expressing Myc-Sqh with Flag, 86.6% cells (Figure 6=149) had no PLA signal and 13.4% of cells with weak PLA signal (≤ 10 foci). By contrast, 82.9% (Figure 6=216) of S2 cells expressing both Flag-Vib and Myc-Sqh showed PLA fluorescence: 15.3% of cells displayed strong PLA signal (>30 foci), 24.5% of cells showed moderate PLA signal (11-30 foci) and 43.1% of cells showed weak PLA signal (<10 foci). On average, controls Flag-Vib with Myc and Myc-Sqh with Flag resulted in 1.7 and 0.3 PLA foci per cell, respectively. By contrast, co-expressing Flag-Vib and Myc-Sqh resulted in 15.2 PLA foci per cell. These data (subsection “Vib interacts with Sqh in vitroand in vivoin BiFC assays”) reinforced that Vib and Sqh physically interact.

Taken together, we conclude that Vib physically associates with Sqh.

[Editors' note: further revisions were requested prior to acceptance, as described below.]

Major issues:The pleiotrophy of the vib phenotype and its involvement in asymmetric cell division and neuroblast self-renewal remain unexplained. The newly added data only partially explain why vib mutant clones can both have ectopic neuroblasts but also lose neuroblasts, especially in light of the observed cellular phenotypes (see also below). Also, vib mutant's cytokinesis defect obscures the interpretation. We suggest to use the over-expression of Vib[T63E) in vib^133/j5A6^ neuroblasts as the starting point. If Vib[T63E] indeed rescues the cytokinesis defect as they claimed, they should be able to more cleanly figure out the true polarity defects that lead to either ectopic neuroblasts or premature loss of neuroblasts in both type I and II lineages.

Following the reviewer’s suggestion, we analysed asymmetric protein localization and segregation in *vib^133/j5A6^* neuroblasts expressing VibT63. As shown in our previous manuscript, Mira was delocalized to the spindle in these metaphase and telophase neuroblasts (we have included the quantification of Mira localization in telophase neuroblasts in Figure 4; subsection “The lipid binding and transfer activity of Vib is critical for asymmetric division and homeostasis of neuroblasts”). Next, in *vib^133/j5A6^* neuroblasts overexpressing VibT63E, 60.6% (n=33) of metaphase neuroblasts still displayed cytoplasmic Pros localization, similar to the delocalization of Pros in *vib^133/j5A6^*metaphase neuroblasts (Figure 4—figure supplement 2.3%, n=45; subsection “The lipid binding and transfer activity of Vib is critical for asymmetric division and homeostasis of neuroblasts”). At telophase, 52.6% of *vib^133/j5A6^*neuroblasts expressing VibT63E showed Pros delocalization at the central spindle (Figure 4—figure supplement 2=21; subsection “The lipid binding and transfer activity of Vib is critical for asymmetric division and homeostasis of neuroblasts”), undistinguishable from *vib^133/j5A6^*telophase neuroblasts (Figure 4—figure supplement 2.8%, n=31, subsection “The lipid binding and transfer activity of Vib is critical for asymmetric division and homeostasis of neuroblasts”). Third, the ectopic neuroblasts seen in both type I and type II neuroblast lineages, as well as the depletion of neuroblasts observed in *vib^133^* MARCM clones are very similar to those observed in *vib^133^* clones overexpressing VibT63E (Figure 4—figure supplement 1 and Figure 4—figure supplement 2; subsection “The lipid binding and transfer activity of Vib is critical for asymmetric division and homeostasis of neuroblasts”).

Taken together, these new data indicate that asymmetric division defects clearly contributed to the neuroblast self-renewal phenotypes observed in *vib^-^* neuroblasts, regardless of cytokinesis defects.

It is not entirely convincing that premature nuclear localization of Prospero is the cause of neuroblast loss in vib mutant clones. First, the level of nuclear Prospero shown in Figure 1—figure complement 2B is so much lower than the Prospero+ cells in the same clone. Second, the authors acknowledged that type II neuroblasts do not express Prospero. These data strongly suggest that yet-to-be-identified factors contribute to premature differentiation of vib neuroblasts. If this is the case, the authors should openly acknowledge this possibility, and tone down their interpretation.

We show that nuclear Prospero localization was also observed in interphase *vib^133/j5A6^*neuroblasts expressing VibT63E (Figure 4—figure supplement 2; subsection “The lipid binding and transfer activity of Vib is critical for asymmetric division and homeostasis of neuroblasts”). However, given that the nuclear Pros observed in *vib^-^* neuroblasts was very weak, we cannot exclude the possibility that other yet-to-be-identified factors are responsible for the premature differentiation of *vib^-^*neuroblasts. We have toned down our claim on the contribution of premature nuclear localization of Prospero on the premature differentiation of *vib^-^* neuroblasts (subsection “Vib controls homeostasis of larval central brain neuroblasts”).

Third, it seems that polarity is largely restored in telophase – which should actually be more carefully quantified and displayed in a main figure panel – and thus it is difficult to reconcile how neuroblast homeostasis is affected.

The ratio of segregated apical/basal proteins determines whether cells adopt neuroblast or GMC fate following the neuroblast division (Carbernard and Doe, 2009). Although in *vib^-^* telophase neuroblasts, Mira/Pros/Brat was still observed at the basal side of the cortex, the amount of basal cell fate determinants eventually segregated into basal daughter cell was likely reduced due to their mislocalization on the spindle, which can potentially result in the formation of ectopic neuroblasts. We have already included the quantification of telophase localization of both aPKC and Mira in *vib^1105/j5A6^*and *vib^133/j5A6^*telophase neuroblasts in our previous manuscript (Figure 2 and Figure 2—figure supplement 1). As described above, we have now included the quantification of Mira and Pros in *vib^133/j5A6^*metaphase and telophase neuroblasts expressing VibT63E (Figure 4 and Figure 4—figure supplement 2; subsection “The lipid binding and transfer activity of Vib is critical for asymmetric division and homeostasis of neuroblasts”).

Specific comments:The authors have tried to replace the word "neuroblast overgrowth" with "ectopic neuroblasts" but missed several. Please make the following corrections for consistency. Figure 4—figure supplement 1 and the figure legend associated with it.

Done.

Introduction – The following statements should be modified: "At telophase, nonmuscle Myosin II localizes to the cleavage furrow and regulates basal protein localization (Barros et al., 2003). "The non-muscle myosin II regulatory light chain protein Spaghetti-squash (Sqh) regulates both asymmetric localization of basal cell fate determinants in neuroblasts and cytokinesis of dividing cells (Karess et al., 1991; Barros et al., 2003)." It is true that in sqh germ line clones, basal cell fate determinants are mislocalized. Similarly, the authors show here that both aPKC and Mira fail to localized to the apical and basal neuroblast cortex, respectively in sqh RNAi expressing neuroblasts. However, the authors should also state that Miranda's basal confinement is due aPKC mediated phosphorylation as shown by Atwood et al.

We have modified our statements to “Asymmetric localization of Mira was thought to be achieved via a series of linear inhibitory regulations from aPKC to Mira via Lethal giant larvae (Lgl) and nonmuscle Myosin II (Kalmes et al., 1996; Barros et al., 2003; Betschinger et al., 2003). aPKC was shown later to directly phosphorylate both Numb and Mira to polarize them in neuroblasts (Smith et al., 2007; Atwood and Prehoda, 2009), while Lgl directly inhibits aPKC in neuroblasts, rather than displacing Mira on the cortex (Atwood and Prehoda, 2009).” (Introduction).

Throughout the manuscript the authors should thus make a distinction between "anchoring" of polarity proteins to the cell cortex and "asymmetric localization" of the investigated polarity proteins. It seems to me, that the polarity proteins cannot be localized asymmetrically without the proper anchoring. Lack of Sqh and Vib seems to predominantly affect the anchoring but will not reveal new insight into the mechanism for polarized localization and subsequent asymmetric segregation of these proteins. Thus, the phenotype is more general and Vib and phosphatidolinositol lipids might not necessarily regulate asymmetric cell division specifically.

We have carefully used the terms of “anchoring” and “asymmetric localization” throughout the revised manuscript. We have also included the following statement in the Discussion: “Given the lipid-binding and transfer function of Vib, it is conceivable that it anchors Sqh to the cell cortex prior to the asymmetric localization of Sqh and basal proteins in neuroblasts, rather than directly regulating asymmetric localization of Sqh or basal proteins.” (Discussion section).

Considering recent publications (Roubinet, 2017; Montembault, 2017), the following statement should also be updated or changed: "However, how myosin localization is regulated during asymmetric division remains unknown".

We have changed the statement to: “However, how myosin regulates asymmetric division of neuroblasts remains not well understood.” (Introduction).

The authors claim that Vib is localized to the cell cortex with a basal enrichment. I am not convinced that the basal enrichment is specific for Vib; Figure 1—figure supplement 1 and E shows that Vib is also enriched on the apical side. It seems that Vib enrichment mostly occurs whenever other cells are in contact with the imaged neuroblast. In our experience, membrane markers tend to show basal enrichment due to GMCs clustered on the basal neuroblast side. Thus, if the authors want to show that basal enrichment is specific, they should generate primary neuroblast cultures expressing Vib::Venus of neuroblasts devoid of any attached GMCs.

We have included the following statement in the manuscript: “The slight enrichment of VenusVib at the basal side was likely due to the attachment of multiple GMCs surrounding the neuroblasts, which is commonly seen for membrane proteins.” (subsection “Vib controls homeostasis of larval central brain neuroblasts”).

The authors conclude that Vib is critical for asymmetric protein localization and segregation in dividing neuroblasts. I think this statement needs to be modified. I agree that the localization of polarity proteins is affected in metaphase. However, all provided images of the basal cell fate determinant Miranda show normal localization and segregation in telophase (despite some Mira signal on the spindle). Thus, I am confused why the authors conclude that Vib is critical for asymmetric protein segregation if Miranda and aPKC are normally localized in many vib mutant telophase neuroblasts.

The ratio of segregated apical/basal proteins determines whether cells adopt neuroblast or GMC fate following the neuroblast division (Carbernard and Doe, 2009). Although in *vib^-^* telophase neuroblasts, Mira/Pros/Brat was still observed at the basal side of the cortex, the amount of basal cell fate determinants eventually segregated into basal daughter cell was likely reduced due to their mislocalization on the spindle, which can potentially result in the formation of ectopic neuroblasts. Given the partial delocalization of Mira/Pros/Brat in telophase neuroblasts, we have toned down our claim to “Vib is required for asymmetric protein localization and their faithful segregation in dividing neuroblasts” (subsection “Vib is required for asymmetric division of neuroblasts”).

The following statement "the localization of Sqh::GFP at cleavage furrow was lost" and the subsequent conclusions should be modified. Figure 3 shows that in vib mutant telophase neuroblasts, Sqh::GFP is still localized at the neuroblast cortex (especially with the 1105 allele) although a pronounced furrow enrichment is lacking. It seems as if the overall distribution of Sqh is changed compared to wild type. However, the provided image still shows it to be localized at the cell cortex.

We have modified our statement to “24.2% of telophase neuroblasts had increased cytoplasmic localization of Sqh::GFP (Figure 3=33).” (subsection “Vib anchors non-muscle myosin II regulatory light chain Spaghetti-squash (Sqh) to the cell cortex in neuroblasts”).

I still don't understand the nature of vib133's pleiotropic phenotype. Vib133 mutant clones either show an increase in neuroblast number, lose neuroblasts and have cytokinesis defects. I appreciate the new data showing that expression of Pros could lead to differentiation of vib mutant neuroblasts. However, how can that phenotype be explained considering the observed polarity defects? Loss of aPKC usually causes loss of neuroblasts and not an increase in Nb numbers.

The ectopic neuroblast phenotype in *vib^-^* brains is likely caused by the mislocalization of multiple basal proteins (including cell fate determinants) Mira, Pros and Brat on the spindle as well as spindle misorientation (Figure 2, Figure 1—figure supplement 2 and Figure 4—figure supplement 2; subsection “Vib is required for asymmetric division of neuroblasts”). Loss of aPKC and gain of weak nuclear Pros in neuroblasts as well as other unidentified factors may contribute to the loss of neuroblasts observed in *vib^-^* mutants (subsections “Vib controls homeostasis of larval central brain neuroblasts” and “Vib is required for asymmetric division of neuroblasts “).

Similarly, I am unconvinced that the observed spindle orientation phenotype could result in ectopic neuroblasts; to this end, the authors should show telophase figures, demonstrating that misaligned metaphase spindles indeed give rise to symmetric neuroblast divisions, producing siblings with neuroblast fate. Given the relatively weak nature of the spindle orientation defects, I am not convinced that the authors could find this.

Indeed, we have observed orthogonal division of *vib^133/j5A6^* neuroblasts with an increased sample size (Figure 2; 2.3%, n=86). This new data suggests that defective spindle orientation can potentially contribute to the altered sibling cell fate observed in *vib^-^*mutants. We have updated the manuscript (subsection “Vib is required for asymmetric division of neuroblasts”) and Figure 2.

I am not an expert on the lipid strip assay, however, I feel that a positive control is missing in this experiment (Figure 5).

Lipid strip assay described in our manuscript was carried out with full length Bazooka (Baz) as a positive control. The result for Baz (please refer to Author response image 1) was similar to what was reported in Krahn et al., 2010 (Figure 4). Although in the previous reported paper, full-length Baz was not tested in the lipid strip assay, the strong binding of Baz truncated forms to PtdIns(4,5)P_2_ and PtdIns(3,4,5)P3 was consistent with our results using full length Baz. Therefore, our lipid strip assay is working.

I am further confused by the role of PI(4)P in cortical Sqh localization. If PI(4)P is essential for cortical Sqh localization – which the authors claim is mostly lost in vib mutants – why is the localization of PI(4) in vib mutants only affected at the cleavage furrow in vib mutants? Overall, neuroblasts still contain robust PI(4)P localization in vib mutants. The same is also true for sqh mutants, which still contain robust levels of PI(4)P at the neuroblast membrane.

The reviewer seems to be confused about the reciprocal regulations between Vib/Sqh and PI(4)P in neuroblasts. We conclude that both Vib and PI4KIIIα anchor Sqh to the neuroblast cortex, because Sqh cortical localization is compromised upon loss of Vib or PI4KIIIα Figure 3 and Figure 7). Given that Vib transfers π to PI4KIIIα for the synthesis of PI(4)P and the binding of Sqh with PI(4)P (Figure 5), PI(4)P is likely required for the anchoring of Sqh on neuroblast cortex. On the flip side, we also found that both Vib (Figure 5) and Sqh facilitate the membrane localization of PI(4)P, as cytoplasmic aggregates of PI(4)P-GFP were observed in neuroblasts upon depletion of *vib* or *sqh* (Figure 5). Since PI(4)P was still observed on the plasma membrane, in addition to the cytoplasmic aggregates, in *vib^-^*and *sqh^-^* mutant neuroblasts, we conclude that both Vib and Sqh facilitate the membrane localization of PI(4)P in neuroblasts (subsection “Sqh binds to phosphoinositide PI(4)P and localizes it to cell cortex in neuroblasts”).

[Editors' note: the author responses to the re-review follow.]

All reviewers have agreed that your work provides important new insights into asymmetric division of the Drosophila neuroblasts and thus merits publication in eLife.However, before we can formally accept your paper, we would like you to make following textual edits. After this modification, we will be able to accept your manuscript editorially.1) The reviewers felt that your explanation of 'the ratio of segregated apical vs. basal fate determinant' was not satisfying, especially because no data are offered to back up this notion. Thus, as of now, this explanation is hand-waving. Although the authors do not need to add more experiments and quantification (unless it is easily possible by analyzing existing data), the authors should acknowledge this shortcoming in interpreting their data.

We have modified our statement to “Given that Mira/Pros/Brat was still observed at the basal side of the cortex, it is unclear whether the formation of ectopic neuroblasts is partially contributed by the reduced amount of basal determinants eventually segregated into basal daughter cell due to their mislocalization on the spindle.” (subsection “Vib is required for asymmetric division of neuroblasts”).

2) Also, now that the authors do not claim that Vib is concentrated on basal side and state that Vib is (early) evenly cortical, it clearly leaves the readers wondering what Vib might be doing at the apical side. The reviewer acknowledges that testing 'localization-specific function' is difficult for anybody to do, and do not ask the authors to conduct such experiments: however, again, the authors should more carefully discuss about this, and at least propose possible models.

We have included the following statement in the Discussion section “In addition, Vib’s localization in neuroblast cortex lacks apparent asymmetry, while it is required for the asymmetric localization of apical proteins aPKC and Par-6, but not Baz. We speculate that Vib might also play a role in membrane localization of aPKC or Par-6 in neuroblasts.” (Discussion section).

3) Also, the authors claim that "We have already included the quantification of telophase localization of both aPKC and Mira in vib^1105/j5A6^ and vib^133/j5A6^ telophase neuroblasts in our previous manuscript (Figure 2 and Figure 2—figure supplement 1)".I can find the telophase data for Mira and Pros in Figure 4 and Figure 4—figure supplement 2 but not for aPKC. Maybe it was present in an earlier version and was since removed. If so, it would make sense to include it again.

In the revised manuscript, we have shown the aPKC localization in metaphase and telophase *vib^133/j5A6^*neuroblasts and *vib^133/j5A6^*overexpressing VibT63E in Figure 4—figure supplement 1 and subsection “The lipid binding and transfer activity of Vib is critical for asymmetric division and homeostasis of neuroblasts”, Figure 4—figure supplement 1 legend.